# Thermodynamically inconsistent extreme precipitation sensitivities across continents driven by cloud-radiative effects

Sarosh Alam Ghausi [1,2,3] ✉, Erwin Zehe [3], Subimal Ghosh[4,5], Yinglin Tian[6] & Axel Kleidon [1]

Extreme precipitation events are projected to intensify with global warming, threatening ecosystems and amplifying flood risks. However, observation-based estimates of extreme precipitation-temperature (EP-T) sensitivities show systematic spatio-temporal variability, with predominantly negative sensitivities across warmer regions. Here, we attribute this variability to confounding cloud radiative effects, which cool surfaces during rainfall, introducing covariation between rainfall and temperature beyond temperature's effect on atmospheric moisture-holding capacity. We remove this effect using a thermodynamically constrained surface-energy balance, and find positive EP-T sensitivities across continents, consistent with theoretical arguments. Median EP-T sensitivities across observations shift from −4.9%/°C to 6.1%/°C in the tropics and −0.5%/°C to 2.8%/°C in mid-latitudes. Regional variability in estimated sensitivities is reduced by more than 40% in tropics and about 30% in mid and high latitudes. Our findings imply that projected intensification of extreme rainfall with temperature is consistent with observations across continents, after confounding radiative effect of clouds is accounted for.

Extreme precipitation events can lead to catastrophic floods and are expected to intensify with global warming following the increase in atmospheric moisture at the rate of 7%/K (Clausius-clapeyron (CC) rate)[1,2]. While the climate models have been largely able to simulate such an increase[3–6], observational evidence to validate these changes is challenged by diverging trends and the inhomogeneity of rainfall records[7]. Alternatively, studies test this response in observations by estimating extreme precipitation sensitivities with respect to local near-surface air temperatures, forming an equivalent relationship to the Clausius-Clapeyron equation[8–11]. These so-called "extreme precipitation-temperature (EP-T) scaling rates" show significant deviations from the CC rate of 7%/K with systematic zonal variations.

EP-T scaling shows a monotonic increasing relationship at high latitudes, "hook" like structures in mid-latitudes, and a monotonic decreasing EP-T relationship in the tropics[12–16]. Consequently, EP-T sensitivities remain largely negative over most of the tropical regions in contrast to positive changes projected by climate models and observed trends[7,17–19].

A number of factors have been argued to cause deviations in the EP-T scaling, including shifts in atmospheric dynamics[3], atmospheric stability[20], change in rainfall types from stratiform to convective[21], rainfall event duration[22,23], cooling effect of rain[24–26], availability of moisture[27] and saturation deficits at high temperatures[28]. Alternative scaling variables, like atmospheric and upper tropospheric air

[1]Biospheric Theory and Modelling Group, Max Planck Institute for Biogeochemistry, Jena, Germany. [2]International Max Planck Research School for Global Biogeochemical Cycles (IMPRS – gBGC), Jena, Germany. [3]Institute of Water Resources and River Basin Management, Department of Civil Engineering, Geo and Environmental Sciences, Karlsruhe Institute of Technology – KIT, Karlsruhe, Germany. [4]Department of Civil Engineering, Indian Institute of Technology Bombay, Mumbai, India. [5]Interdisciplinary Programme in Climate Studies, Indian Institute of Technology Bombay, Mumbai, India. [6]Department of Earth System Analysis, Potsdam Institute for Climate Impact Research (PIK) – Member of the Leibniz Association, Potsdam, Germany. ✉e-mail: sghausi@bgc-jena.mpg.de

temperatures[29,30], and moisture indicators such as dew point temperature and integrated water vapor[31–33], have been proposed but offer only marginal enhancements in scaling. Using dew point temperatures showed improvements[34], but it underestimates rainfall-depth information[35], fails to fully address negative scaling and peak structures in tropical regions[16,36,37], and provides limited insights on rainfall sensitivity to increasing temperatures[26,38]. As a result, there remain large uncertainties in using present-day scaling to project changes in future precipitation extremes.

Here, we show that a large part of this uncertainty in EP-T scaling can be explained by the confounding radiative effect of precipitating clouds on surface temperature. Clouds associated with rainfall events significantly alter the local surface energy budget as they reflect shortwave radiation to space while they absorb and re-emit longwave radiation back to the surface. Consequently, they cause changes in surface temperatures mostly resulting in a net cooling across warmer regions and periods. This cooling makes observed temperatures covary with precipitation and thereby affects the causal nature of precipitation-temperature scaling relationships. As a result, scaling rates not only show how precipitation changes with temperature but also reflect how the synoptic conditions associated with the rainfall event affect temperature. The primary objective of this study is to quantify to what extent the covariation between clouds and temperatures affects the regional EP-T sensitivities.

We evaluate this effect by combining observationally derived daily gridded rainfall and temperature datasets from the Climate Prediction Center (CPC) and Global Precipitation Climatology Project (GPCP – 1.3)[39] with satellite-based daily observations of cloud-area fraction and radiative fluxes from the NASA-CERES dataset[40,41]. The consistency of results was also checked with ERA-5 reanalysis data. To remove the effect of clouds on surface temperatures, we used a surface energy balance model where the vertical turbulent exchange is explicitly constrained by the thermodynamic limit of maximum power[42,43]. By using this additional constraint together with the "all-sky" and "clear-sky" radiative fluxes as forcings, we estimate changes in surface temperatures associated with cloud radiative effects during rainfall

events. Subsequently, we use them to evaluate the impact of clouds on rainfall-temperature scaling.

## Results and discussion

### Observed scaling of extreme precipitation with temperature

We start by estimating the extreme precipitation-temperature (EP-T) scaling rates across the global land grids using the quantile regression (QR) method[9]. It involves fitting a QR model between the logarithmic precipitation and temperature values at the target quantile of 95%, 99%, and 99.9%. The EP-T sensitivity (scaling rate) is then calculated by the exponential transformation of the slope coefficient (see "Methods"). Figure 1A shows the map of estimated sensitivities from the CPC observations for the 95th percentile[13,36], along with their zonal variations (Fig. 1B). Positive sensitivities were found at high latitudes, while in the tropics, the sensitivities remained mostly negative. The sensitivities were particularly negative in the tropical humid regions of India, Nothern Australia, Central Africa, Western US, and Amazonia (Fig.1A). These patterns were consistent across different rainfall quantiles, including the 99th and 99.9th percentiles, and for the GPCP and ERA-5 reanalysis data as well (Supplementary Figs. S1, S9, and S10). Sensitivities estimated using ERA-5 data were generally more negative than those derived from CPC and GPCP rainfall observations (Supplementary Fig. S10).

To further look at distinct patterns in EP-T relationships, we use LOWESS (locally weighted scatter plot smoothing regression) for the 95th percentile daily rainfall events (hereafter P95) and corresponding near-surface air temperature at each grid cell[12] and identified three distinct scaling behaviors. This includes a monotonic decreasing (MD) relationship, a hook-shaped (HS) relationship, and a monotonic increasing (MI) relationship. The MD and HS characteristics were predominant in the tropics and mid-latitudes, while the MI relationship was mainly found over grids at high latitudes. Figure 1C–E illustrates the EP-T scaling for all the grids (depicted by light orange lines) in the tropics, mid-latitudes, and high-latitudes respectively. Most of the grid cells in the tropics showed a monotonic decreasing relationship. In mid-latitudes, most of the grid cells showed "peak structures" such

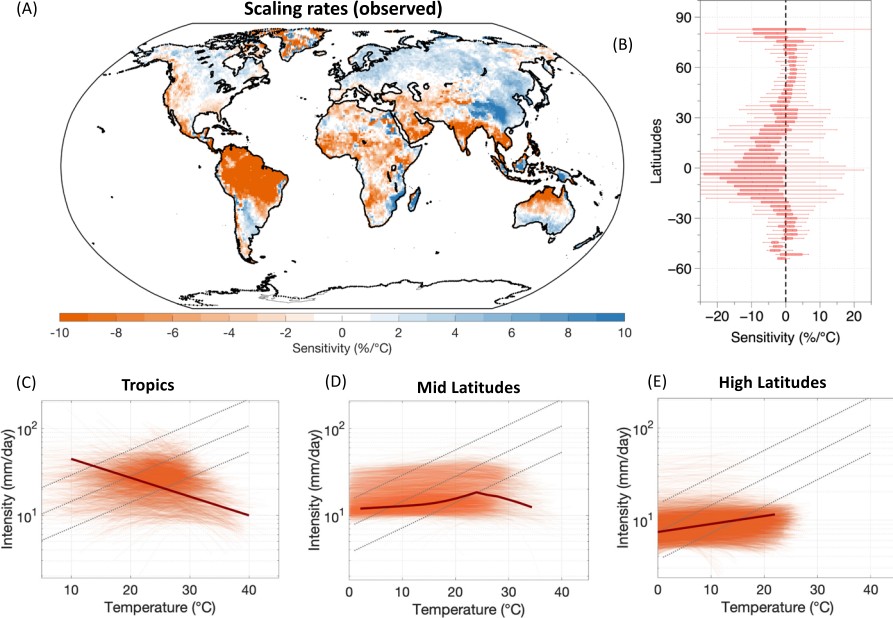

**Fig. 1 | Scaling of extreme precipitation with temperature in observation. A** Global map of extreme precipitation (P95) – observed temperature (EP-T) sensitivities estimated from quantile regression method, (**B**) zonal variation of estimated EP-T sensitivities. Scaling curves between daily extreme rainfall intensity and temperature for (**C**) tropics, (**D**) mid-latitudes, and (**E**) high-latitudes. The light orange lines show the scaling curves for each grid while the dark orange line indicates the mean response of all the grid cells. The black dotted line indicates the CC rate (7%/°C). Note the logarithmic y-axes in (**C**–**E**).

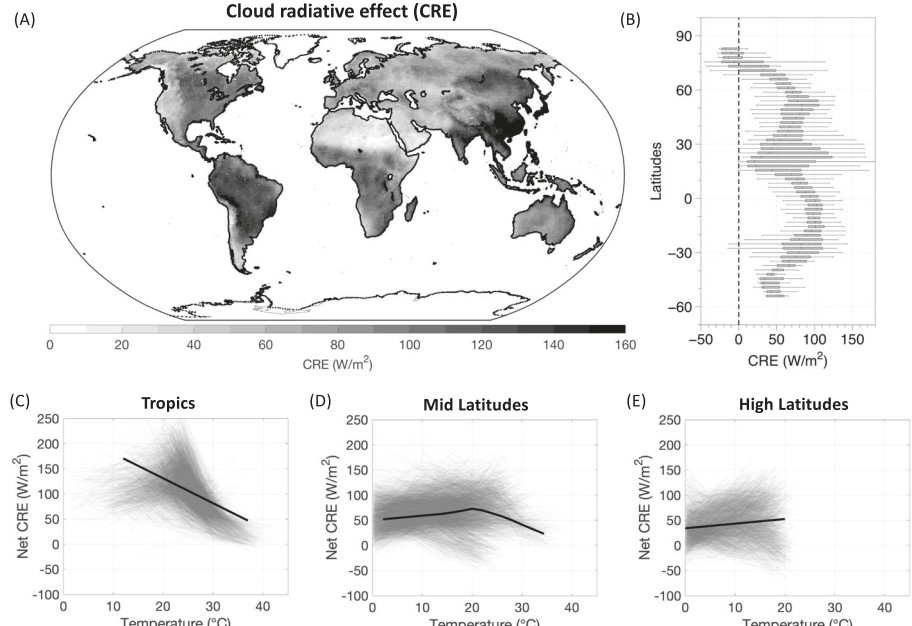

**Fig. 2 | Scaling of cloud-radiative effects with temperatures. A** Global map of net cloud radiative effect (CRE) defined as the difference between "clear-sky" and "all-sky" radiative fluxes including both shortwave and longwave radiation, isolated on the days when rainfall is greater than P95 (95th percentile). **B** zonal variation of estimated CRE. Scaling curves between CRE and observed temperatures for (**C**) tropics, (**D**) Mid-latitudes, and (**E**) High-latitudes. The gray lines show the scaling curves for each grid cell while the black line indicates the mean response of all grid cells.

that extreme precipitation increases with temperature up to a certain threshold of around 20 °C, after which the scaling breaks down and the EP-T relationship becomes negative. The high latitudes showed a monotonic increasing relationship but the rate of increase remained lower than CC rates. Note that the tropics were characterized as regions from 23S to 23 N, mid-latitudes as regions from 23 N – 55 N and 23S – 55S, and high-latitudes as regions beyond 55S and 55 N.

The negative sensitivities in the warmer tropics, systematic zonal variation in observed sensitivities and the three distinct characteristics shown by EP-T scaling as depicted in Fig.1 are consistent with what has already been reported by previous studies[12,15,16,24,37].

**Observed scaling of cloud radiative effects with temperatures**

We next show that these distinct patterns in EP-T scaling can be reproduced alone by how the radiative effect of clouds during rainfall events affects temperatures. To illustrate this, we calculated the cloud radiative effects (CRE), which we defined as the difference between "clear-sky" and "all-sky" radiative fluxes. The positive values implies a reduction in radiative heating at the surface by clouds, indicating cooling, while the negative values indicate warming of the surface. We estimated the CRE for both shortwave and downwelling longwave radiation during the extreme rainfall (P95) events. The global distribution of shortwave and longwave CRE are shown in Supplementary Fig. S2. When plotted as a function of rainfall (Supplementary Fig. S3), we found that the shortwave CRE shows a statistically significant increase as the rainfall intensity increases leading to a reduction of more than 100 W/m² of energy from reaching the Earth's surface and thereby exerting a strong cooling effect. The strength of CRE was proportional to the intensity of the rainfall event. The CRE of downwelling longwave radiation had an opposite warming effect, but the magnitude remain much lower in comparison to shortwave CRE. It further remains largely insensitive to changes in rainfall (Supplementary Fig. S3). This insensitivity can be partly explained by the compensating effects of emissivity and atmospheric heat storage in shaping downwelling longwave radiation[44]. The net effect of clouds on the surface is then diagnosed by adding the CRE of both shortwave and longwave radiation and is referred to as the net CRE. The global

distribution of the net CRE is shown in Fig. 2A. It shows a systematic spatial and zonal variability across the globe such that the humid tropical regions where the averaged extreme rainfall can exceed 40 mm/day are associated with a very large CRE of more than 120 W/m² , whereas in the dry regions, the heavy rainfall is largely limited by available moisture and show a reduced CRE of less than 40 W/m² (Supplementary Fig. S2). It also shows a systematic zonal variation with higher CRE in the tropics compared to higher latitudes (Fig. 2B).

We then generated the scaling curves for radiation-temperature scaling (hereafter "CRE-T scaling") curves equivalent to EP-T scaling (Fig. 2C−E). We find that the tropical regions showed a monotonic decreasing relationship for CRE-T scaling. Mid-latitudes showed a "hook-structure" in CRE-T scaling as well, while high latitudes showed a monotonic increasing CRE-T scaling. These curves are similar to EP-T scaling but have quite a different interpretation. The relationships of CRE and temperature clearly reflect that temperatures covary with extreme rainfall events and are lowered during the strongest events primarily because of the reduction in radiative heating of the surface due to clouds. As a result of this cooling, the strongest rainfall events are shifted to lower temperature bins, causing the bin-shifting effect − where the apparent sensitivity of precipitation to temperature gets biased by the redistribution of extreme events into cooler bins. This effect leads to an overestimation of precipitation sensitivity at cooler temperatures and an underestimation at warmer temperatures, skewing the perceived precipitation-temperature relationship[24,26]. This effect is strongest in the tropics and at the higher temperature bins in the mid-latitudes where the slope of CRE-T scaling is negative. This shows that the three distinct EP-T scaling curves shown in Fig. 1C−E do not only indicate how the rainfall events change with temperature but also reflect the covariation induced by how the radiative conditions associated with these events affect temperatures.

To quantify how much the changes in CRE affect surface temperatures, we use a thermodynamically constrained surface energy balance model, forced it with radiative fluxes for "all-sky" and "clear-sky" conditions, and estimated "all-sky" and "clear-sky" temperatures that include and excludes the radiative effects of clouds respectively (see details in Ghausi et al. [26] & methods section)[26]. This model

captures the observed day-to-day variation of land surface temperature derived from NASA-CERES reasonably well with the $R^2$ of 0.96 and mean RMSE of 3.8 K (Supplementary Fig. S4). The sensitivity of surface temperatures to changes in cloud cover was also very well captured with an $R^2$ of 0.93 (Supplementary Fig. S5). The residual errors between the model and observations were comparable and only slightly larger than those between observations and ERA-5 reanalysis data (Supplementary Figs. S4C, S5B). The cooling effect of clouds given by $\Delta T_{clouds}$ was then quantified as the difference between clear-sky and all-sky temperatures. This temperature difference ($\Delta T_{clouds}$) varies as a function of precipitation (Supplementary Fig. S6) with the heaviest rainfall associated with the strongest amount of cooling. However, at very high rainfall levels, this effect tends to saturate, which is likely due to a saturation in the cloud-area fraction and associated radiative effects. The global variation of this cooling effect is shown in Supplementary Fig. S7. We found that the humid tropical regions experience the most significant cooling and the effect dampens as we move towards drier regions and higher latitudes. The reduction in cooling at high latitudes is because of an increasing compensating role of enhanced downward longwave radiation compared to the reduction in absorbed solar radiation, while over dry regions it is simply because of the lack of clouds.

## Extreme precipitation-temperature scaling corrected for cloud radiative effects

We then evaluated the changes in the EP-T scaling rates and their variations after the cloud effects on temperatures were removed. The adjustment to temperatures was made by adding the estimated temperature difference by clouds ($\Delta T_{clouds}$) to the observations of air temperature during rainy days (see "Methods"). The global distribution of precipitation temperature scaling rates for the 95th percentile rainfall events, estimated after adjusting for the cloud-cooling effects, is shown in Fig. 3A. We found positive sensitivities over most of the global land regions. The zonal variation showed positive values across the latitudinal range (Fig. 3B). These results were consistent across different rainfall quantiles, including the 99th and 99.9th percentiles, and for the GPCP and ERA-5 reanalysis data as well (Supplementary

Figs. S8–S10). The breakdown in scaling at high temperatures also disappeared in the scaling curves, and rainfall extremes showed a monotonic increase closely aligning with the CC rate of 7%/°C throughout the temperature range (Fig. 3C–E). The most significant change in scaling was found in the tropics where the radiative effects of clouds are the strongest. Diametric change in sensitivities from negative to positive was found over the tropical humid regions of India, Southeast Asia, Nothern Australia, Central Africa, and Amazonia. These positive sensitivities are consistent with the regional climate model projections and observed trends over these regions[24,45,46]. Notably, some grids in Eastern China showed positive EP-T sensitivities for both observed and clear-sky scaling (Figs. 1A, 3A), despite experiencing a strong cloud-radiative effect (Fig.2A). We believe that this effect is likely related to the different regimes of how cloud-radiative effects vary with increasing rainfall and the seasonality of monsoon-rainfall and its effect on pre-monsoon temperatures (see Supplementary Fig. S12 for more detail). Other studies have reported a similar spatial pattern in observed scaling over Eastern China as well[16,47]. Different scaling between the Himalaya Mountain regions and southeast Asia can also be attributed to topographic changes that have been shown to affect rainfall dynamics and scaling patterns[48]. The clear-sky scaling also showed a variation with aridity such the scaling rates remain lower/negative across the dry regions. This can be seen over the regions of the Middle East Asia and Western United States that still showed negative scaling (Fig. 3A and Supplementary Fig. S16), We note that this pattern cannot be explained by the effect of clouds alone and may likely relate to the moisture-availability limitations[27]. The mid-latitudes exhibited weaker scaling at low temperatures but showed a sharp increase at higher temperatures closely aligning with CC rates. This increase in scaling between 10 °C to 20 °C is consistent with other studies and has been argued to be a result of changing rainfall type from stratiform rainfall at low temperatures to convective rainfall at high temperatures[21,49]. The scaling over high latitudes remained positive and largely unchanged due to weak cloud radiative effects. Similar results were obtained when the analysis was repeated with the GPCP rainfall data (Supplementary Fig. S8). These findings suggest that the argued intensification of extreme rainfall with temperatures is

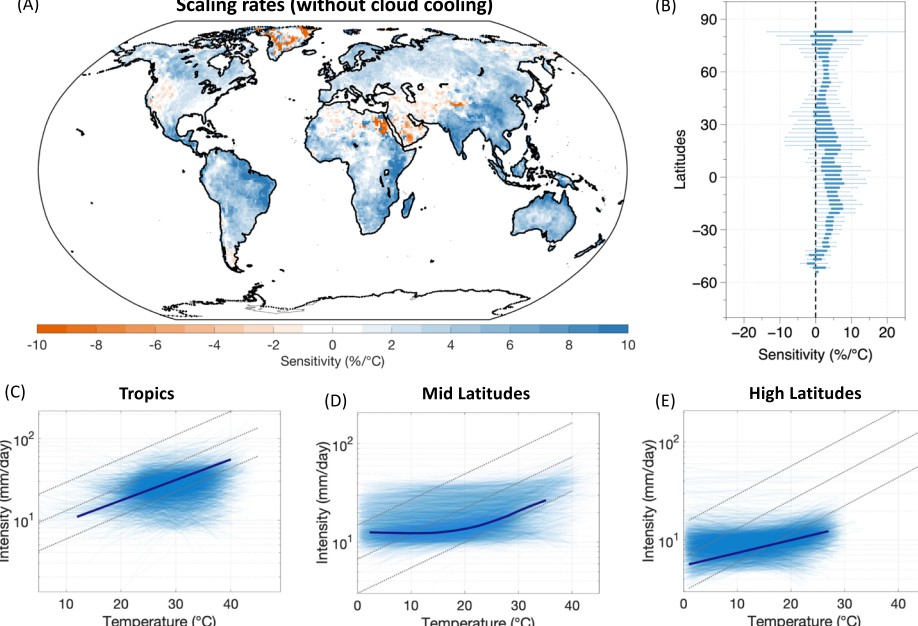

**Fig. 3 | Scaling of extreme precipitation with temperature after correcting for cloud radiative effects. A** Global map of extreme precipitation (P95) – temperature sensitivities (EP-T) without the effect of cloud cooling, (**B**) zonal variation of estimated sensitivities. Scaling curves between daily extreme rainfall intensity and temperature (cloud-adjusted) for (**C**) tropics, (**D**) Mid-latitudes, and (**E**) High-latitudes. The light blue lines show the scaling curves for each grid cell, while the solid dark blue line indicates the mean response of all the grid cells. The black dotted line indicates the CC rate (7%/°C). Note the logarithmic y-axes in (**C–E**).

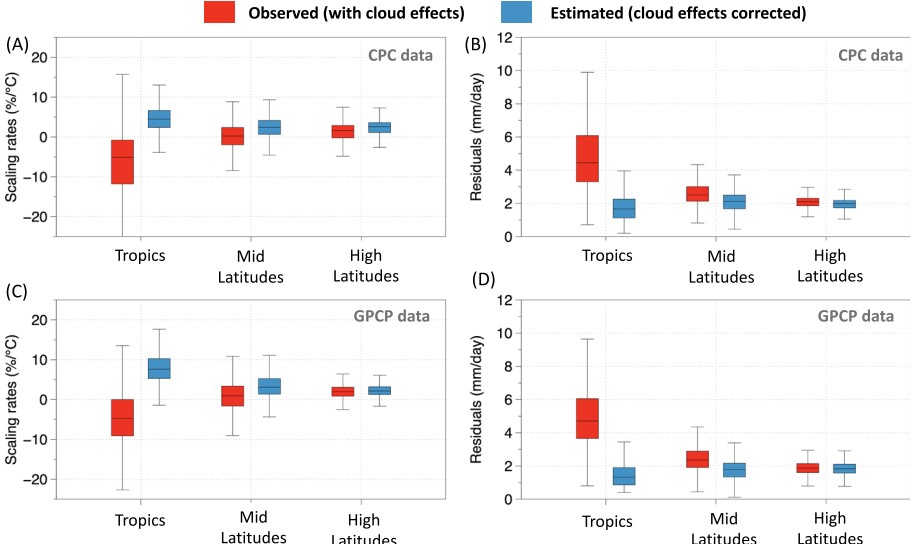

**Fig. 4 | Comparison of extreme precipitation-temperature scaling rates with and without cloud-cooling correction across regions and datasets.** Extreme precipitation-temperature (EP-T) scaling rates were estimated using observed temperatures (red) and with temperatures corrected for the cloud-cooling effect (blue) for tropics, mid-latitudes, and high-latitudes for (**A**) CPC data and (**C**) GPCP data. **B** and **D** same as (**A** and **C**) but for the residuals between observations and fitted quantile regression.

consistent with observations across global land after the covariation between clouds and temperatures is corrected.

Figure 4A then closely compares the observed and derived scaling rates with and without the effect of clouds, respectively. Results show that after removing the cloud-temperature covariation, the median scaling rates in the tropics changed from − 5.1%/°C to 4.5%/°C in the CPC data (Fig. 4A) and from − 4.7%/°C to 7.6%/°C in the GPCP data (Fig. 4C). The change in mid-latitudes were from − 2%/°C to 2.4%/°C in the CPC data (Fig. 4A) and 0.9%/°C to 3.1%/°C in GPCP data (Fig. 4C). The variability in the estimates of scaling rates was also substantially reduced. The interqauartilrange (IQR) of the estimated sensitivities was reduced by about 40% in the tropics, while the total range was reduced by more than 50%, where the cloud radiative effects are the strongest (Fig. 4B). The IQR was reduced to about 30% in mid and high latitudes. Similar reductions were also observed using GPCP rainfall data (Fig. 4D). The reduction in variability was statistically significant for all three zones, as determined by an F-test ($p < 0.0001$), where p represents the probability of obtaining the observed difference in variances under the null hypothesis of equal variances. In addition, we diagnosed the residuals between the observations and fitted quantile regression. The residuals consistently showed lower values for estimates where the cloud effects on temperatures were removed across the latitudinal range, with the most significant reduction occurring in the tropics (Fig. 4B) for both CPC and GPCP data. This then relates back to our interpretation that the primary source of uncertainty in observed EP-T response arises from the covariation of rainfall and temperature through the confounding radiative effect of clouds.

There still exists regional variability and deviations from the CC rate in the estimated "clear-sky" sensitivities over land. Studies attribute these regional deviations to dynamic factors, particularly to changes in vertical pressure velocities[11,50]. To assess the impact of dynamics on regional variations in EP-T sensitivities, we analyzed standardized anomalies of vertical pressure velocity (w) on extreme rainfall days, using ERA-5 data at 650 hPa. The results reveal a positive correlation such that stronger anomalies in vertical velocity lead to positive deviations from the CC rate, particularly between 30 N and 30 S, where dynamic effects have been shown to be more pronounced[11] (Supplementary Fig. S14). These changes in vertical velocities are tightly coupled to updrafts within the clouds, where greater power

from condensation heating drives deeper convection and increased rainfall. We further explored this by examining the difference between cloud-base and cloud-top temperature as a proxy for moist convection and found a clearer, monotonic increasing relationship with deviations in EP-T sensitivities (Supplementary Fig. S14B, D). This highlights the role of dynamics in regulating regional variability in EP-T sensitivities, consistent with previous research[11,50].

Most negative EP-T scaling also occurs over tropical oceans[16,51]. To investigate the impact of cloud radiative cooling on these scaling estimates, we extended our analysis to include oceans using rainfall-temperature data from ERA-5 reanalysis and radiative fluxes from NASA-CERES. Our energy-balance model was able to reproduce sea-surface temperatures across oceans reasonably well (Supplementary Fig. S12A, B). The scaling became positive across the tropical oceans after the cloud effects were removed (Supplementary Fig. S12C, D). These estimates were consistent with what has been reported using dew-point temperatures as a scaling variable over oceans[16]. These results confirm that the negative scaling in oceans can be explained in parts by cloud-radiative effects as well. However, it is to be noted that cooling over oceans may also result from factors like wind-induced upwelling, turbulent mixing, and heat extraction by storms which remain unaccounted for in our current approach.

We found that the number of grids exhibiting sensitivities greater than the Clausius-Clapeyron rate (super-CC scaling) over land increases more than threefold, after removing the cloud radiative effects from temperature (Supplementary Fig. S13). Most of this change occurred in the tropics, with regional patterns being consistent across observations (CPC) as well as ERA-5 reanalysis data (Supplementary Fig. S13). The super CC scaling can not be explained by the thermodynamic constraints on saturation vapor pressure alone and has been attributed to the dynamic factors including enhanced moisture convergence and increased moist updrafts within clouds[8,52]. These results suggest that the presence of super-CC scaling in observed EP-T sensitivities is largely underestimated when cloud effects on temperatures are not taken into account.

In the mid and high latitudes, the scaling rates were positive and showed a monotonic increase with temperatures but remained less than the CC rate across most grids. While the monotonic increase has already been attributed to the predominance of thermodynamic

factors shaping EP-T sensitivities over these regions, the negative deviations from CC rate over these grids can be due to many other factors that have been shown to cause discrepancies in observed EP-T scaling and are not accounted for here. This primarily includes consideration of rainfall-event duration[22,23], changing rainfall types[21], shifts in atmospheric dynamics[50], the role of topography[48], and lack of moisture availability[27]. While these factors also affect scaling, our finding shows that the systematic zonal variation with negative tropical sensitivities and breakdown in scaling at high temperatures can be largely explained by accounting for the cloud radiative effects alone. It is crucial to clarify that the objective of this study is not to explain all the variability in EP − T scaling but to quantify to what extent the covariation of rainfall and temperature alone affects these estimates. Our results revealed that after removing the cloud radiative effects, the EP-T sensitivities yielded physically consistent estimates and showed a significant reduction in their spatial variability.

The role of the cooling effect of rainfall in impacting the EP-T scaling has also been argued before[24,25,28,37]. However, no study has explicitly attributed this effect to the observed zonal variation in the global scaling rates. Part of it is due to the complexities associated with quantifying this cooling effect across the global scale, particularly over regions where we lack rainfall-temperature data at high temporal resolution. To correct this effect, atmospheric moisture measures like dew-point temperature[31] have been suggested as an alternative scaling variable to dry bulb temperatures. Dew-point temperature showed improved scaling, but they do not entirely account for confounding synoptic changes within the atmosphere[38]. It was shown that while the effect of clouds may not be directly visible in extreme precipitation - dew point scaling, the breakdown occurs in how dew-point scales with observed temperatures[26,33,38]. Time-lagged daily temperatures have also been used to minimize the cooling effect[24]. While they show some improvement in scaling, they do not entirely remove the negative sensitivities. We show in Supplementary Fig. S15 that explicitly correcting for cloud effects using radiation data, clearly outperforms time-lagged temperature estimates of EP-T scaling. Other methodologies to correct for the cooling effect of rainfall on temperatures had been largely statistical, relying on the very fine temporal resolution of temperature and precipitation data[25,37], which may not be available in all the regions. While we agree that the use of fine-resolution data may improve the scaling estimates[53], it remains an emerging challenge, particularly for future projections in data-scarce regions. On the contrary, our physics-based approach to remove the cloud cooling effect relies only on globally available satellite observations of radiative fluxes and provides an efficient tool to correct this effect across the globe and provide physically consistent estimates of global and regional EP-T sensitivities.

Our results imply that the "hook structures" and high-temperature thresholds at which the scaling breakdown or negative scaling occurs (also called "peak temperatures"), will not limit the intensification of extreme rainfall in a future climate. This is consistent with the reported shift in peak temperatures towards warmer and wetter conditions diagnosed using climate model projections[51,54,55]. However, the extent to which the modeled shift in peak temperatures relates to cloud-radiative effects in climate model projections remains a potential area for future research.

Our findings can also be extended to address the observed inconsistency between extreme rainfall and streamflow sensitivities to increase in temperatures[13,15,56]. This inconsistency has largely been attributed to factors such as rainfall duration and antecedent soil moisture conditions[15,57], however the role of clouds in affecting these differences has not been explored yet. Given that the time of concentration can vary significantly across catchment sizes, the temperatures sampled for rainfall and streamflow extremes may correspond to different radiative conditions influenced by cloud effects, potentially playing a key role in affecting these sensitivities.

Our approach, which removes the confounding effect of clouds on temperatures, thus provides an efficient tool for diagnosing the sensitivities of other hydrological variables beyond rainfall extremes as well.

To conclude, our analysis demonstrates that the observations across global land are consistent with the argued intensification of rainfall extremes with global warming after the covariation between clouds and temperature is accounted for. Cloud radiative effects can explain most of the observed variability in the extreme precipitation-temperature sensitivities including negative scaling rates in the tropics and the breakdown in scaling at high temperatures. This resolves the discrepancy between the apparent negative scaling rates in observations and the projected increase in precipitation extremes by climate models. More work is still needed to understand the existing uncertainties about observed changes in the rainfall dynamics with global warming in order to reliably extrapolate observationally derived EP-T sensitivities directly into the future[3,11]. Our findings further highlight that the rising threat of increased rainfall extremes with global warming across continents is already evident in observations, emphasizing the urgent need for more effective strategies for climate adaptation and disaster preparedness.

## Methods
### Datasets used
We used daily rainfall data from CPC – global daily precipitation dataset[58] which is available at 0.5° x 0.5° resolution. This dataset is derived using gauge-based observations across the globe. The results were also validated using the daily rainfall data from the Global Precipitation Climatology Project (GPCP – 1.3)[39] available at 1° x 1° resolution. This dataset is combined using satellite-based products and in situ observations. Observed gridded temperature data was used from the CPC-global daily temperature dataset available at 0.5° x 0.5° resolution. The analysis was also repeated using rainfall-temperature data from the ERA-5 reanalysis interpolated at 1° x 1° resolution. Surface and top-of-atmosphere radiative fluxes data for "all-sky" and "clear-sky" conditions were obtained from NASA CERES – Syn1deg dataset[40,41] available at 1° x 1° resolution. The analysis was performed on a daily scale over the years from 2001 to 2023.

### Physical model behind scaling
The scaling of extreme rainfall with temperature is widely adopted to understand extreme precipitation-temperature sensitivities across regions and has been extensively reviewed[11,59,60]. The physical equation arises using the vertically integrated dry static energy budget (derived in Muller et al.[61]) and can be expressed as:

$$P_e = \epsilon \int \rho w \left( -\frac{\partial q_{sat}}{\partial z} \right) dz \qquad (1)$$

Where $P_e$ is the rate of extreme precipitation, $\epsilon$ denotes the precipitation efficiency such that $\epsilon = 1$ indicates that all condensation precipitates out. $\rho$ denotes the mean density, w is the vertical velocity and $q_{sat}$ is the saturation mixing ratio. Assuming small changes in vertical velocities and precipitation efficiency with warming, the fractional change in rainfall can then be written as

$$\frac{\delta P_e}{P_e} \approx \frac{\delta q_s(T_{surface})}{q_s(T_{surface})} \qquad (2)$$

This indicates that the changes in extreme rainfall can be approximated to follow changes in saturation vapor pressure ($q_s$) at the surface which is an exponential function of surface temperatures (Clausus-Clapeyron equation). As a result, one can isolate extreme rainfall events and analyze their rate of change in observations using the exponential relationship with local surface temperatures[8,9,13].

## Estimation of scaling rates

EP-T sensitivities were estimated by using the quantile regression (QR) method, which calculates the conditional quantile of the dependent variable (in this case, precipitation) based on given values of the independent variable (temperature). This methodology to estimate EP-T sensitivities has been widely adopted by previous studies[8,9,12]. The first step involved fitting a QR model between the logarithmic precipitation and temperature values at the target quantile (n) (95%,99%, and 99.9% in our case)

$$Log(P_i) = \beta_o^n + \beta_1^n(T_i) \tag{3}$$

Here $P_i$ denotes the daily rainfall intensity and $T_i$ is the observed daily temperature, and $\beta_o^n$ and $\beta_1^n$ are the regression coefficients for the $n^{th}$ quantile. To estimate the rate of increase, the slope coefficient $\beta_1^n$ is exponentially transformed and is referred to as the scaling rate ($\alpha_1$).

$$\alpha_1(\%/°C) = 100.(e^{\beta_1^n} - 1) \tag{4}$$

## Quantifying the cloud radiative effects on surface temperatures

To remove the effect of cloud cooling from surface temperatures, we used a thermodynamically constrained surface energy balance model. The thermodynamic constraint arises by setting the vertical turbulent exchange to operate at the thermodynamic limit of maximum power. This approach has already been evaluated against observations and showed excellent agreement in reproducing turbulent fluxes and surface temperatures[43,62,63]. We force this model with absorbed solar radiation, downwelling longwave radiation at the surface, and outgoing longwave radiation at the top of the atmosphere for both "all-sky" and "clear-sky" conditions. These fluxes are a standard product in the NASA-CERES dataset, where 'all-sky' fluxes reflect the observed conditions, including cloud radiative effects while "clear-sky" fluxes are derived by eliminating the cloud effects from the radiative transfer. The temperature difference associated with clouds was estimated by the difference in the modeled temperature for "clear-sky" and "all-sky" conditions. This difference was then added to the observed temperatures during rainy days in order to eliminate the cloud-cooling effect. A detailed description of this approach can be found in Ghausi et al.[26] and briefly below.

## Thermodynamic constraint on turbulent flux exchange

The vertical turbulent flux exchange was conceptualized as an outcome of a heat engine operating between the two reservoirs – the hot surface and the cooler atmosphere. The surface is heated by absorbed solar radiation ($R_s$) and downwelling longwave radiation ($R_{ld}$) which makes it warmer at temperature ($T_s$). This energy is released back to the atmosphere but at a much colder temperature ($T_r$) described by outgoing longwave radiation at the top of the atmosphere ($R_{l,toa}$). This heat engine performs work to sustain the convective motion. The work done by this engine is proportional to the temperature difference between the two reservoirs. At the same time, the more work the engine performs, it leads to more vertical turbulent exchange between surface and atmosphere, which in turn depletes the driving temperature difference. This flux-gradient feedback then leads to an optimal turbulent flux that maximizes the total work done. This limit is termed as the maximum power limit[42].

After writing the first and second law of thermodynamics for this conceptualized heat engine (see details in ref. 26 and ref. 43), the total power generated by this heat engine can be written as

$$G = \left(J_{in} - \frac{dU}{dt}\right)\left(\frac{\left(\frac{R_s + R_{ld} - J_{in}}{\sigma}\right)^{\frac{1}{4}}}{T_r} - 1\right) \tag{5}$$

Here, $J_{in}$ is the vertical turbulent flux exchange, dU/dt denotes the heat storage changes within the heat engine, and σ is the Stefan – Boltzmann constant with the value of $5.67 * 10^{-8}$ Wm$^{-2}$ K$^{-4}$. The optimal turbulent flux ($J_{opt}$) that maximizes the power generation (G) was estimated by numerically maximizing Eq. (5) with respect to $J_{in}$. This approach has already been evaluated against observations and has shown excellent agreement in reproducing turbulent fluxes and surface temperatures[43,62,64,65]. Additional evaluation is shown in Supplementary Figs. S4, S5.

After estimating the optimal turbulent fluxes $J_{opt}$, the surface temperatures at the maximum power limit can be calculated by using the surface energy balance as described in Eq. (6).

$$T_{max\,power} = \left(\frac{R_s + R_{ld} - J_{opt}}{\sigma}\right)^{\frac{1}{4}} \tag{6}$$

To estimate the effect of clouds on surface temperatures, the maximum power model is forced with the radiative fluxes for "all-sky" and "clear-sky" conditions. We use these fluxes to estimate "all-sky" and "clear-sky" surface temperatures using Eqs. (7) and (8).

$$T_{all\,sky} = \left(\frac{R_{s,all\,sky} + R_{ld,all\,sky} - J_{opt,all\,sky}}{\sigma}\right)^{\frac{1}{4}} \tag{7}$$

$$T_{clear\,sky} = \left(\frac{R_{s,clear\,sky} + R_{ld,clear\,sky} - J_{opt,clear\,sky}}{\sigma}\right)^{\frac{1}{4}} \tag{8}$$

The effect of clouds on surface temperatures was then calculated as the difference between "all-sky" and "clear-sky" temperatures as

$$\Delta T_{clouds} = T_{clear\,sky} - T_{all\,sky} \tag{9}$$

Finally, we apply this correction to the Eq. (2) to estimate changes in extreme precipitation as

$$\frac{\delta P_e}{P_e} \approx \frac{\delta q_{sat}(T_{surface} + \Delta T_{clouds})}{q_{sat}(T_{surface} + \Delta T_{clouds})} \tag{10}$$

## Data availability

All the datasets used in this study are freely accessible. CPC Global Unified Precipitation data was provided by the NOAA/OAR/ESRL PSD, Boulder, Colorado, USA, and can be accessed from their Web site at https://psl.noaa.gov/data/gridded/data.cpc.globalprecip.html. Global Precipitation Climatology Project (GPCP) Climate Data Record (CDR), Version 1.3 (Daily) is freely accessible from https://doi.org/10.5065/ZGJD-9B02. CPC Global Unified Temperature data was provided by the NOAA PSL, Boulder, Colorado, USA, and can be accessed from their website at https://psl.noaa.gov/data/gridded/data.cpc.globaltemp.html. Surface and TOA gridded radiative fluxes for all-sky and clear-sky conditions were obtained from NASA-CERES Syn1deg data (https://doi.org/10.5067/Terra+Aqua/CERES/SYN1degDay_L3.004A, NASA Langley Research Center, Atmospheric Science Data Center, 2021). ERA-5 reanalysis data can be accessed from https://doi.org/10.24381/cds.adbb2d47 and https://doi.org/10.24381/cds.bd0915c6. Raw data files for each figure can be assessed from https://doi.org/10.5281/zenodo.11449685[64].

## Code availability

The code to run the maximum power model can be accessed at https://doi.org/10.5281/zenodo.11449685. The Matlab code to apply the quantile regression can be assessed from https://www.mathworks.com/matlabcentral/fileexchange/32115-quantreg-x-y-tau-order-nboot.

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

## Acknowledgements

We thank the NASA-CERES team for providing open access to the data, the Copernicus Climate Change Service for granting access to the ERA-5 reanalysis data. S.A.G., E.Z., and A.K. acknowledge financial support from the Volkswagen Stiftung through the ViTamins project. S.A.G and A.K. acknowledge funding from the Max Planck Institute for Biogeochemistry, Jena-Germany.

## Author contributions

S.A.G. led the conceptualization, methodology development, investigation, interpretation, and writing of the original draft. A.K. contributed to the conceptualization, methodology development, interpretation, and review/editing. E.Z. contributed to the conceptualization, interpretation, and review/editing. S.G. contributed to the conceptualization, interpretation, and review/editing. Y.T. contributed to the interpretation and review/editing.

## Funding

## Competing interests

The authors declare no competing interests.
