## [Transparent Peer Review file · Nature Communications]

Thermodynamically inconsistent extreme precipitation sensitivities across continents driven by cloud-radiative effects

Corresponding Author: Dr Sarosh Ghausi

Version 0:

Reviewer comments:

Reviewer #1

(Remarks to the Author)

Review Report of NCMMS-24-33469

Extreme precipitation is projected to intensify with global warming. However, estimated extreme precipitation-temperature sensitivities exhibit negative behaviors in tropics and a 'hook' structure in mid-latitudes. Previous studies have reported various reasons causing these deviations from the Clausius-Clapeyron relationship: changes in rainfall types, varying duration of rainfall events, availability of moisture, cooling effect of rain, etc. This manuscript explored the behind mechanisms over global land areas and reveals that negative sensitivities in warm periods and regions are primarily results from confounding cloud radiative effects on surface temperatures during the rainfall events. This topic regarding investigating why 'negative precipitation-temperature' relationships is interesting; the authors do provide a strong KEY finding contribute to improve the understanding of extreme precipitation physics. The language is fluent and the whole storyline is coherent. However, I still have some concerns that need to be addressed before the article being accepted at such a high-impact journal.

MAJOR COMMENTS:

1. The authors used a thermodynamically constrained surface-energy balance approach to remove the effect of clouds on temperature; and they found median EP-T sensitivities shifted from negative to positive; the mid-latitudes saw an increase in the sensitivities. This is really interesting. However, existing studies also mentioned that atmospheric dynamics (e.g., vertical velocity) play dominant roles in regulating deviations of extreme precipitation sensitivity from CC relationships and is the main driver of spatial variability (e.g., Pfahl et al., 2017; Gu et al., 2023). Can the authors further explore how the clouds affect the atmospheric dynamics of the extreme precipitation sensitivities?

Pfahl, S., O'Gorman, P. A., & Fischer, E. M. (2017). Understanding the regional pattern of projected future changes in extreme precipitation. *Nature Climate Change*, 7(6), 423-427.

Gu, L., Yin, J., Gentine, P., Wang, H. M., Slater, L. J., Sullivan, S. C., ... & Guo, S. (2023). Large anomalies in future extreme precipitation sensitivity driven by atmospheric dynamics. *Nature Communications*, 14(1), 3197.

2. The authors investigate the observed scaling rates (before removing the effects from clouds) over the global land areas tend to be all smaller than 7%/K (Fig. 1 and Fig. S1); which is kind of strange, previous studies also reported that there is super CC scaling for precipitation-temperature relationships (e.g., Yin et al., 2018; Ali et al., 2022) ; Can the authors explain why in their studies there is only negative-sub CC scaling?

Yin, J., Gentine, P., Zhou, S., Sullivan, S. C., Wang, R., Zhang, Y., & Guo, S. (2018). Large increase in global storm runoff extremes driven by climate and anthropogenic changes. *Nature communications*, 9(1), 4389.

Ali, H., Fowler, H. J., Pritchard, D., Lenderink, G., Blenkinsop, S., & Lewis, E. (2022). Towards quantifying the uncertainty in estimating observed scaling rates. *Geophysical Research Letters*, 49(12), e2022GL099138.

3. The observed scaling rates is negative over the India while positive over the southeast Asia (Fig. 1). However, both regions show strong net cloud radiative effect (Fig. 2). Then after removing the net cloud radiative effect, the scaling rates

over the India become strongly positive while the scaling rates over the southeast Asia almost stay unchanged (Fig. 3). Why this happens? I mean, why the strong cloud radiation effect cannot affect the scaling over the southeast Asia? If they do, the scaling rates after adjustment should also sharply increase. I notice the authors say that the cloud effects dampen towards drier regions and higher latitudes (e.g., western North America). This makes sense since the CRE (cloud radiation effect) is weak there; but since the CRE is really strong over the southeast Asia, why it cannot affect the scaling rates? Please explain more details.

4. The authors explore the regimes behind negative precipitation-temperature scaling relationships over global land areas. This is no doubt meaningful. However, actually, most negative scaling also emerges over tropical oceans (e.g., Wang et al., 2017). Can the authors also extend to discuss whether the cloud effects can explain the negative scaling over tropical oceans?

Wang, G., Wang, D., Trenberth, K. E., Erfanian, A., Yu, M., Bosilovich, M. G., & Parr, D. T. (2017). The peak structure and future changes of the relationships between extreme precipitation and temperature. *Nature Climate Change*, 7(4), 268-274.

5. My final concern is that, the analysis is performed over 2001-2016, only involving 17 years data. The common analysis involves 20-30 years data; since short-term analysis can involve large uncertainty from internal climate variability. To make their results more robust, can the authors extend the analysis period? Or at least can they use some model simulations (e.g., CMIP6 model outputs) to provide further evidence of their results?

SPECIFIC COMMENTS:

1. Line 30: consistent with model projections. This sounds weird. Please change to 'consistent with observed trends' or something like that. The model projections are not always tell the truth. And the adjusted results are consistent with model projections cannot prove the explanation is reasonable.

2. Line 139-140: the humid tropical regions are associated with CRE of more than 120 W/m² whereas the dry regions show a reduced CRE of less than 40 W/m². How the authors define humid tropical regions? where the dry regions refer to? The authors should define these regions at least in the supplementary.

3. Line 211-214, the median scaling rates in the tropics changed from -3%/k to 5%/k. The change in mid-latitudes from 0.8%/K to 3.2%/K. Again, the authors need to clarify these regions either in the methods section or in the supplementary.

4. In Figure 4, again, I feel strange, why observed scaling rates over tropics, mid-latitudes, high-latitudes both show negative to sub-CC scaling rates? This is inconsistent with previous findings. Please clarify this.

5. The authors only use one extreme quantile, 95%, to exhibit their findings. To make the results more robust, I suggest they perform the analysis using varying extreme quantiles, e.g., 99%, 99.9%.

(Remarks on code availability)

I have no further comment regarding the code.

Reviewer #2

(Remarks to the Author)

This study explores the climate-scale cloud radiative effects on surface temperature during rainfall and quantifies their impacts through a thermodynamically constrained surface energy balance model. The study is an extension of the authors' previous works about estimate of heat flux based on a thermodynamically constrained energy balance model, and the cloud-radiative effect on surface temperatures and turbulent fluxes over land (Ghausi et al. 2023). The major results suggest that the projected intensification of extreme rainfall with temperature in tropics is closer to the ideal pattern of Clausius-Clapeyron rate when the cloud radiative effects are removed. While the overall results do provide certain evidence to support the explanation of the inconsistent relationship between temperature increase and extreme rainfall increase in observations, we have concerns about applying the thermodynamically constrained energy balance model to daily data at finer grid scale to seek relation between surface temperature (with and without cloud effect) and extreme rainfall (P95). Although the authors claim that the approach have already been evaluated against observations and showed excellent agreement in reproducing turbulent fluxes and surface temperatures (Kleidon & Renner 2017; Ghausi et al., 2023), it appears that these evaluations are done based on data at longer time scale not at daily time scale. If true, there is no guarantee that the approach works on daily time scale because the simple energy balance may not hold in local convective processes for extreme rainfall. This is shown in the highly scattered EP-T relation in Fig. 3. In fact, there is no physical base to expect EP-T to follow thermodynamic relation. There are few studies to suggest the P-T relation slower than the q-T relation in global energy balance condition, but no existing theories explain the EP-T relation. Even the q-T relation varies in different climate regimes (e.g., super greenhouse effect over the tropical warm pool). Therefore, I would expect to see some justification or evaluation of the thermodynamically constrained energy balance approach. Such an evaluation requires an estimate of quality of daily variables of solar and terrestrial radiative fluxes at surface, the surface temperature (that is different from near surface air temperature), and turbulent fluxes.

A related question is about the change of water cycle with environment warming. For completeness, it is desirable to see the same analysis over oceans. The SST, air-sea fluxes, and rainfall are all available over oceans. It is physically more suitable and practically easier to apply the method to search EP-T relation over oceans.

Besides the above major concerns, we wonder whether the cloud cooling effect is the most important factor influencing EP-T. The authors mentioned some other explanations in the text. Is it possible to isolate the EP-T under greenhouse warming in the current climate?

Overall, the objectives and findings of this study are quite interesting but the applicability of the method needs be justified to ensure the major conclusions are reasonable. In addition, some statistics, physical discussions, and method descriptions are unclear and must be revised. Therefore, I recommend that this article be considered for publication after the above concerns are adequately addressed.

Specific comments:

1. The extreme precipitation-temperature (EP-T) sensitivities, scaling rate, and quantile regression should be briefly introduced/defined in the main text, although they are described in the section of Data and Methods.
2. Please add more physical discussion about possible mechanisms causing the different effects among the tropics, mid-, and high-latitudes.
3. Please highlight the potential applications of the findings in this study or the expansion of our physical understanding in the last 2-3 paragraphs. They are unclear in the current manuscript.
4. The "significant results" (e.g., lines 129, 244, 274, Fig 4, Fig S3, S4, S7) should be examined by a statistical significance test to precisely mention and compare their significant differences or regressions.
5. Please add a brief description of "the bin-shifting effect" in line 152 to the main text or an example figure in the supplement.
6. Line 199: Fig S5 should be Fig S6.
7. CPC should be in full name when it first appears.
8. Each possible mechanism mentioned in lines 235-237 should have corresponding references or brief descriptions.
9. Please ensure that either °C or K is used consistently throughout the manuscript.
10. Lines 240-242: It should be also highlighted in the Introduction.
11. Line 246: has
12. Line 260: remains
13. Line 265: suggest
14. Line 272: highlight
15. Line 274: show
16. Line 283: induced
17. Line 285 shows
18. Please check and add many missing references, such as Emory & Brown., 2005; Lenderink et al., 2010; Dhara et al., 2016; Kleidon & Renner 2017; Wasko et al., 2018; Wasko & Nathan 2019; Tian et al., 2023a, b ...etc.

(Remarks on code availability)

The manuscript is subject to a major revision that will require a substantial amount of time and effort. This may be a factor in deciding "major revision" or "reject"

Reviewer #3

(Remarks to the Author)

(Remarks on code availability)

Reviewer #4

(Remarks to the Author)

The observed scaling relationship between surface temperature and extreme precipitation daily amount is revisited at global scale. The temperature is corrected for, thanks to a surface budget equation, to remove the cooling effect of the cloudiness associated with the extreme precipitation event. The resulting correction improves the scaling, in particular in the tropics.

The authors should be commended for connecting the cloud radiative effect of the precipitating system to the surface temperature in the scaling relationship. This is a timely addition to the debate on the usefulness of such scaling laws for

assessing climate models predictions of future extreme precipitation.

While this reviewer is rather in favor of publication of this paper, it will increase the impact of this work if the paper would include a discussion about:

- physical model behind the scaling (e.g. Muller and Takayabu, 2020) and how the corrected temperature actually enters the scaling equation?
- alternative to this correction. In particular using time lagged temperature instead of contemporaneous ones over both land (Bao et al., 2017) and ocean (De Meyer and Roca, 2021) and how the present correction does a better job at characterizing the surface condition that yield to the extreme precipitation event?

On a miscellaneous note, I would suggest to state early in the manuscript the source of the precip data and the scale (daily) as well as the scale upon which the cloud radiative effect is computed.

References

- Bao, J., Sherwood, S., Alexander, L. et al. Future increases in extreme precipitation exceed observed scaling rates. *Nature Clim Change* 7, 128–132 (2017). <https://doi.org/10.1038/nclimate3201>
- De Meyer, V. and R. Roca, 2021: Thermodynamic scaling of extreme daily precipitation over the tropical ocean from satellite observations. *J. Meteor. Soc. Japan*, 99, 423-436.
- Muller, C. J., & Takayabu, Y. (2020). Response of precipitation extremes to warming: what have we learned from theory and idealized cloud-resolving simulations, and what remains to be learned? *Environmental Research Letters*. <https://doi.org/10.1088/1748-9326/ab7130>

(Remarks on code availability)

Version 1:

Reviewer comments:

Reviewer #1

(Remarks to the Author)

1. The authors address most of mine concerns, this is good. However, I still have some doubts regarding the scaling patterns across India and southeast China. The authors argue that 'different regimes of how CRE scales with rainfall may likely relate to the contribution from large scale and convective rainfall events.' Since it is tropics, I would like to believe large scale circulation plays a minor role while small scale circulation and convection can be more important. Moreover, is this phenomenon can be related to 'monsoons'? However, even during monsoon seasons, strong precipitation across the India and southeast China can be the same... Can the authors discuss or analyze more details about how the monsoon affect the different scaling across India, southeast China, and Indonesia, respectively?

(<https://scied.ucar.edu/learning-zone/storms/monsoons>)

2. Secondly, I still consider that in the main text, dividing all scaling patterns into tropics, mid-latitude, high latitudes are not that appropriate? Maybe it's not a big problem, but since the authors have already global maps, why they always tend to group the results into 3 region boxes? Since there are discernable zonal differences of the scaling rates. Further, does the aridity can affect the scaling rates or cloud effects? For example, how does the observed scaling rates varies and how cloud cooling affects the eastern North America (belongs to humid areas) and the western North America (belongs arid areas), respectively? Finally, it seems that topography can also affect scaling (e.g., different scaling between the Himalaya mountain regions and southeast Asia). Although it is not the target of this research, the authors can have a short discussion.

(Remarks on code availability)

Reviewer #2

(Remarks to the Author)

The revised manuscript addresses all my comments adequately. I recommend the manuscript be accepted for publication after some very minor changes as stated below. I do not need to see the revision again.

1. The descriptions about figure S4ab are unclear and confusing. I do not understand NASA-CERES temperature. From the "dataset used", I see CPC temperature but no NASA-CERES temperature.
2. In LINE 465, equation (S3) is referred but nonexistent. Please check equation numbers in the text.

(Remarks on code availability)

Reviewer #3

(Remarks to the Author)

(Remarks on code availability)

Version 2:

Reviewer comments:

Reviewer #1

(Remarks to the Author)

All my concerns have been well addressed and I consider at this stage it can be accepted.

(Remarks on code availability)

Reviewer #1 (Remarks to the Author):

Review Report of NCMMS-24-33469

Extreme precipitation is projected to intensify with global warming. However, estimated extreme precipitation-temperature sensitivities exhibit negative behaviours in tropics and a 'hook' structure in mid-latitudes. Previous studies have reported various reasons causing these deviations from the Clausius-Clapeyron relationship: changes in rainfall types, varying duration of rainfall events, availability of moisture, cooling effect of rain, etc. This manuscript explored the behind mechanisms over global land areas and reveals that negative sensitivities in warm periods and regions are primarily results from confounding cloud radiative effects on surface temperatures during the rainfall events. This topic regarding investigating why 'negative precipitation-temperature' relationships is interesting; the authors do provide a strong KEY finding contribute to improve the understanding of extreme precipitation physics. The language is fluent and the whole storyline is coherent. However, I still have some concerns that need to be addressed before the article being accepted at such a high-impact journal.

We thank the reviewer for investing their time and effort and for providing a positive and thoughtful assessment of our work. The reviewer suggested important points that needed more analysis and clarifications. We have now accordingly revised the manuscript with the point by point response mentioned in this document. The reviewer comments are shown in Black and our responses are shown in Blue. All the reviewer comments have been marked for reference as R1_M1, R1_M2, R1_M3 respectively. Each reply is composed of a "response" to the comment and "action" taken in the main manuscript/supplementary file.

MAJOR COMMENTS:

R1_M1: The authors used a thermodynamically constrained surface-energy balance approach to remove the effect of clouds on temperature; and they found median EP-T sensitivities shifted from negative to positive; the mid-latitudes saw an increase in the sensitivities. This is really interesting. However, existing studies also mentioned that atmospheric dynamics (e.g., vertical velocity) play dominant roles in regulating deviations of extreme precipitation sensitivity from CC relationships and is the main driver of spatial variability (e.g., Pfahl et al., 2017; Gu et al., 2023). Can the authors further explore how the clouds affect the atmospheric dynamics of the extreme precipitation sensitivities?

Pfahl, S., O’Gorman, P. A., & Fischer, E. M. (2017). Understanding the regional pattern of projected future changes in extreme precipitation. *Nature Climate Change*, 7(6), 423-427.

Gu, L., Yin, J., Gentine, P., Wang, H. M., Slater, L. J., Sullivan, S. C., ... & Guo, S. (2023). Large anomalies in future extreme precipitation sensitivity driven by atmospheric dynamics. *Nature Communications*, 14(1), 3197.

Response:

This is an important point about the spatial variability in EP-T sensitivities. While, we show in figure 4 (main manuscript) that the regional variability in the extreme precipitation-

temperature (EP-T) sensitivities is reduced after removing the cloud radiative effect from temperatures, there still exist regional variations and deviations from CC rate in the estimated “clear-sky” sensitivities. Studies have shown that these deviations arise due to the dynamic component characterized by changes in vertical pressure velocities (Pfahl et al., 2017; Gu et al., 2023).

To test how much changes in dynamics affect the regional variations of EP-T sensitivities in our study, we looked at standardized anomalies of vertical pressure velocity (w) isolated on extreme rainfall days at each grid point. Vertical pressure velocity at 650 hpa were used from ERA-5 reanalysis data. We plotted it against the spatial deviation in the “clear-sky” EP-T sensitivities from CC rates (Fig. R1A). We found an increasing relationship indicating that the stronger anomalies in vertical pressure velocity leads to positive deviations in sensitivities from CC rate while lower values lead to negative deviations. The relationship becomes clearer and stronger when the grids between 30N and 30S were isolated (Fig. R1C). This is the same region where the dynamic effects of changes in vertical velocities have been shown to be predominant (Gu et al., 2023).

These changes in vertical pressure velocities are tightly coupled to updrafts within the clouds. Heat release during condensation generates power within clouds to perform work to generate motion and dehumidify the atmosphere. The greater power the clouds generate, deeper is the convection and higher is the rainfall. To analyse these differences in cloud updrafts, we used the differences between cloud base and cloud top temperatures (obtained from NASA-CERES observations) as a proxy for moist convection within clouds. The higher are the updrafts, deeper the cloud grows and higher is the difference between cloud-base and cloud-top temperatures. We show in Fig. R1 B, D that the spatial deviation in EP-T sensitivities shows a clear monotonic increasing relationship with standardized anomalies in the cloud temperature differences on extreme rainfall days. This relationship is similar and clearer to the relationship found using anomalies in vertical pressure velocities.

This shows that dynamics play an important role in regulating deviations in EP-T sensitivities, consistent with findings of other studies (Pfahl et al., 2017; Gu et al., 2023). However, in order to estimate the magnitude of this effect on EP-T sensitivities, one has to explicitly account for an analytical expression of moist convection within clouds and estimate its rate of change with temperatures. While we believe that this will be a really interesting result, it remains beyond the scope of the current study which is focused on the impact of cloud-radiative effects on the thermodynamic scaling. However, it remains an important area for potential extension of this research.

Action:

Discussion is added in lines 269-282 of the new manuscript. Fig. R1 (below) have been added as a new supplementary figure S14.

Figure R1: Deviation in EP-T sensitivities (P95) from CC rate (7%/K) after correcting for cloud-effects as a function of (A) standardized anomalies in vertical pressure velocity at 650 hpa on extreme rainfall (P95) days and (B) standardized anomalies in the temperature difference between cloud-base and cloud-top on extreme rainfall days (P95). (C, D) same as (A, B) but for grids between 30S and 30N.

R1_M2: The authors investigate the observed scaling rates (before removing the effects from clouds) over the global land areas tend to be all smaller than 7%/K (Fig. 1 and Fig. S1); which is kind of strange, previous studies also reported that there is super CC scaling for precipitation-temperature relationships (e.g., Yin et al., 2018; Ali et al., 2022); Can the authors explain why in their studies there is only negative-sub-CC scaling?

Yin, J., Gentine, P., Zhou, S., Sullivan, S. C., Wang, R., Zhang, Y., & Guo, S. (2018). Large increase in global storm runoff extremes driven by climate and anthropogenic changes. *Nature communications*, 9(1), 4389.

Ali, H., Fowler, H. J., Pritchard, D., Lenderink, G., Blenkinsop, S., & Lewis, E. (2022). Towards quantifying the uncertainty in estimating observed scaling rates. *Geophysical Research Letters*, 49(12), e2022GL099138.

Response:

We thank the reviewer for the comment. The reviewer correctly points out that previous studies have reported super CC scaling (EP-T sensitivities > 7%/K) in the tropics and mid-

latitudes. However, our analysis found only a limited number of grids exhibiting super-CC scaling. We attribute this primarily to the daily time scale of the datasets used.

To clarify this, we have isolated the grids where super-CC scaling is observed and presented them in Fig. R2. The grid is classified to follow super CC scaling when the sensitivity is greater than 7%/K. The left column in Fig. R2 displays grids with super-CC scaling before accounting for the cloud-cooling effects on temperature, using the CPC dataset (A) and ERA-5 (C). The right column shows the grids with super-CC scaling after correcting for cloud-cooling effects, again for both the CPC dataset (B) and ERA-5 (D).

First, we would like to emphasize that our estimates of scaling that includes cloud effects at the daily time scale, are consistent with the findings of previous studies (Utsumi et al., 2018; Yin et al., 2018; Tian et al., 2023). The specific locations where super-CC scaling occurs are consistent across datasets (CPC and ERA-5) and align very well with the previous research (Utsumi et al., 2011). The differences in our scaling estimates, compared to studies that conduct analyses at sub-daily time scales (e.g., Ali et al., 2022), are primarily in the magnitude of scaling. The impact of storm duration on higher scaling rates has been widely discussed (Ghausi et al., 2020). However, it is important to note that the occurrence of negative sensitivities in the tropics is consistent regardless of the temporal scale of analysis (Tian et al., 2023).

Secondly, we observe that the number of grids exhibiting super-CC scaling increased more than threefold after removing the cloud radiative effects from temperature. This pattern is also consistent across both the observations (CPC) and reanalysis (ERA-5) datasets. These results suggest that the presence of super-CC scaling at daily scale is largely underestimated when cloud effects on temperatures are not taken into account.

Action: Text is added in the lines 294-303 of the revised manuscript. Fig. R2 (below) have been added as new supplementary figure S13.

Figure R2: Grids exhibiting super CC scaling (EP-T sensitivity > 7%/K) in observations (A,B) and ERA-5 reanalysis (C,D) with cloud-effects (A,C) and after correcting for cloud-effects (B,D).

R1_M3: The observed scaling rates is negative over the India while positive over the southeast Asia (Fig. 1). However, both regions show strong net cloud radiative effect (Fig. 2). Then after removing the net cloud radiative effect, the scaling rates over the India become strongly positive while the scaling rates over the southeast Asia almost stay unchanged (Fig. 3). Why this happens? I mean, why the strong cloud radiation effect cannot affect the scaling over the southeast Asia? If they do, the scaling rates after adjustment should also sharply increase. I notice the authors say that the cloud effects dampen towards drier regions and higher latitudes (e.g., western North America). This makes sense since the CRE (cloud radiation effect) is weak there; but since the CRE is really strong over the southeast Asia, why it cannot affect the scaling rates? Please explain more details.

Response: This is a very good point. This leads to even broader question why two regions with similar strong cloud radiative effect behave differently to observed EP-T scaling. We looked into this, by isolating the grids over Southeast Asia. Fig. R3 shows the observed scaling (C) with clouds and (E) after correcting for the cloud radiative effects (CRE). The map of net CRE (defined as the sum of shortwave and longwave CRE) isolated over this region and averaged over all extreme rainfall days is shown in panel (D). While the scaling after removing the cloud effects is positive everywhere, the question is why the south east humid region of China (solid black box in the Fig. R3C) shows positive scaling and regions of Thailand, Laos, Vietnam (dotted black box in Fig. R3C) shows negative observed scaling despite having similar CRE (Fig. R3D). It should also be noted that other studies have reported a similar spatial pattern in observed scaling over Southeast Asia as well (Tian et al., 2023; Moghari et al., 2022).

We show that this effect can be explained by the different regimes of how cloud radiative effects varies with rainfall (Fig. R3B and R3F). In the southeast humid region of China, the net CRE increases with rainfall but saturates at high values implying no further change in cooling with increasing rainfall (Fig. R3B). As a result, when extreme rainfall is scaled against observed

temperatures, the observed scaling (red line in Fig. R3A) also flattens at high temperature bins leading to positive sensitivities. This bin-shifting effect even leads to an apparent super-CC scaling at low temperature bins (see also Li et al., 2023), but it disappears after cloud effects are removed. However, in regions of Vietnam/Laos the CRE keeps on increasing with rainfall (Fig. R3F) resulting in a monotonic decreasing scaling of rainfall with temperatures (Fig. R3E) and negative sensitivities.

These different regimes of how CRE scales with rainfall may likely relate to the contribution from large scale and convective rainfall events. Convective rainfall is localized and concentrated with little spatial spread and may not entirely saturate the grid with clouds however more contribution from stratiform rainfall will result in more cloud saturation of the grid. It may also relate to a latitudinal effect of higher solar insolation for grids closer to equator leading to increased reflectance of top of atmosphere solar-radiation by clouds. We find this an important effect that can be further explored using higher resolution regional datasets.

Action: Discussion is added in lines 220-224 of the revised manuscript. Fig. R3 (below) is added as new supplementary figure S11.

Figure R3: Maps of EP-T sensitivities for observations (C), Net cloud radiative effect on extreme rainfall (P95) days (D) and EP-T sensitivities after correcting for cloud-effects (E) isolated over Southeast Asia region. Black solid and black dotted boxes further separate these regions into two parts. (A, E) shows the scaling curves of extreme precipitation with observed temperatures (red line) and temperatures corrected for cloud-effects (blue-line) for grids within black solid box and black dotted box respectively. (B, F) shows the variation of net cloud radiative effect with rainfall for grids within black solid box and black dotted box respectively.

R1_M4: The authors explore the regimes behind negative precipitation-temperature scaling relationships over global land areas. This is no doubt meaningful. However, actually, most negative scaling also emerges over tropical oceans (e.g., Wang et al., 2017). Can the authors also extend to discuss whether the cloud effects can explain the negative scaling over tropical oceans?

Wang, G., Wang, D., Trenberth, K. E., Erfanian, A., Yu, M., Bosilovich, M. G., & Parr, D. T. (2017). The peak structure and future changes of the relationships between extreme precipitation and temperature. *Nature Climate Change*, 7(4), 268-274.

Response:

The reviewer correctly mentions that most negative scaling has also emerged over tropical oceans (Wang et al., 2017; Tian et al., 2023). To test to what extent cloud radiative cooling affect the scaling estimates over ocean, we have now extended our analysis to oceans as well. We used ERA-5 reanalysis rainfall-temperature data and together with the radiative fluxes from NASA-CERES to estimate the EP-T sensitivities with and without the effect of clouds.

Observed EP-T scaling over oceans is shown in Fig. R4C. We find that the negative scaling dominates across the tropical oceans, with regional patterns being consistent with what have been reported by other studies (Wang et al., 2017; Tian et al., 2023; Gu et al., 2023). To estimate the effect of cloud-cooling over oceans, we used our thermodynamically constrained surface energy balance forced with “all-sky” and “clear-sky” radiative fluxes. The evaluation of the estimated surface temperatures with observations from CERES is shown in Fig.R4 A, B. We find that the residual errors were higher than that compared to land. This can be attributed to the surface heat -storage effect within the oceans that remain unaccounted for in our approach (more details in Kleidon & Renner, 2017). However, this will lead to systematic bias which should not affect the difference between “clear-sky” and “all-sky” temperatures which is quantified as cloud-radiative cooling. Thus, our approach can be used to diagnose the effect of clouds on ocean temperatures as well.

EP-T scaling over oceans after removing the effect of cloud are shown in Fig. R4D. The scaling became positive across the tropical oceans. These estimates were consistent with what have been reported using dew-point temperatures as a scaling variable over oceans (Tian et al., 2023). However, it should also be noted that apart from clouds, ocean surface temperatures can also cool due to wind-induced upwelling of deep sea-water, turbulent mixing of the ocean layers and heat extraction by storm during the rainfall event which remain unaccounted for in our approach. So, while we show that the cloud-cooling effects explains negative sensitivities in tropical oceans, we note that the net estimated cooling of surface temperature over oceans may be underestimated by accounting for the effect of clouds alone. A more detailed analysis to quantify the cooling effect of storm over ocean will require a different formulation of surface energy-balance model that explicitly accounts for the heat-storage and wind mixing effects over the ocean.

Action: Text is added in the lines 283-293 of the revised manuscript. Fig. R4 is added as a new supplementary figures S12.

Figure R4: (A) Comparison of estimated daily sea surface temperatures with the energy balance model with temperatures derived from (NASA-CERES). Root mean squared error (RMSE) between daily estimated and observed temperatures from CERES across all the oceanic grids. Global map of extreme precipitation-temperature sensitivities over oceanic grids with (C) observed temperatures and (D) temperatures corrected for cloud-effects.

R1_M5: My final concern is that, the analysis is performed over 2001-2016, only involving 17 years data. The common analysis involves 20-30 years data; since short-term analysis can involve large uncertainty from internal climate variability. To make their results more robust, can the authors extend the analysis period? Or at least can they use some model simulations (e.g., CMIP6 model outputs) to provide further evidence of their results?

Response: We thank the reviewer for the comment. We have now extended the whole analysis to the period from 2001 to 2023 for both CPC and GPCP data. Additionally, we have now also used the ERA-5 data over both land and ocean to check the robustness of the results. The primary limitation that constrains the analysis period is the availability of observed satellite radiative flux data from NASA-CERES which only starts from 2001. All the results in the new analysis (with extended period) are consistent with the previous finding and does not affect our final conclusions. The use of climate models has been kept out of scope for the present study in order to avoid the uncertainties arising with the representation of day-to-day cloud radiative fluxes in climate model projections which may add to the spread and will require a separate evaluation.

Action: Whole analysis is extended for the years (2001 – 2023). All the main and supplementary figures are modified accordingly. New supplementary figure S10 is added.

SPECIFIC COMMENTS:

1. Line 30: consistent with model projections. This sounds weird. Please change to ‘consistent with observed trends’ or something like that. The model projections are not always telling the truth. And the adjusted results are consistent with model projections cannot prove the explanation is reasonable.

Response: Thanks for pointing this out. The suggested change is made.

Action: Text is accordingly modified in the revised manuscript.

2. Line 139-140: the humid tropical regions are associated with CRE of more than 120 W/m² whereas the dry regions show a reduced CRE of less than 40 W/m². How the authors define humid tropical regions? where the dry regions refer to? The authors should define these regions at least in the supplementary.

Response: We apologize for not being clear. We have now added modified the supplementary figure S2 and explicitly isolated the CRE for each region and described how the region is classified. These regions are described in the figure caption.

Action: Figure R5 is added as new supplementary figure S2.

Figure R5: Global map of cloud radiative effects (defined as the difference between the “clear-sky” and “all-sky” radiative fluxes) isolated on extreme rainfall days for absorbed shortwave (A) and downwelling longwave (C) radiation. Box plots for (B) shortwave CRE and (D) downwelling longwave CRE separated into different climate zones. The tropical regions are

classified as grids between 23S and 23N, mid-latitudes as grids from 23S-55S and 23N-55N and high-latitudes as grids beyond 55S and 55N. The classification into humid and arid regions is derived from the Budyko Aridity index, which is calculated as the ratio of mean annual potential evapotranspiration to mean annual precipitation. The region is classified as humid when aridity index is less than 1 and arid when it is greater than 1.

3. Line 211-214, the median scaling rates in the tropics changed from -3%/k to 5%/k. The change in mid-latitudes from 0.8%/K to 3.2%/K. Again, the authors need to clarify these regions either in the methods section or in the supplementary.

Response: Thanks for pointing this out. We refer to tropics as regions from 23S to 23N. The mid-latitudes are characterized as regions from 23N – 55N and 23S – 55S. Regions higher than 55N and 55S are classified as high-latitude regions. We have now added this into the main text.

4. In Figure 4, again, I feel strange, why observed scaling rates over tropics, mid-latitudes, high-latitudes both show negative to sub-CC scaling rates? This is inconsistent with previous findings. Please clarify this.

Addressed in comment R1.M2

5. The authors only use one extreme quantile, 95%, to exhibit their findings. To make the results more robust, I suggest they perform the analysis using varying extreme quantiles, e.g., 99%, 99.9%.

Response: We agree that performing analysis for different quantiles adds to the robustness of the results. We have now extended the analysis to 99th percentile and 99.9th percentile rainfall (shown in figure below) in addition to 95th percentile (shown in main figures). The key findings remain same for all the percentiles.

Action: Text is accordingly modified in the revised manuscript. Fig. R6 (below) is added as new supplementary figure S9.

Figure R6: Global map of extreme precipitation – temperature (EP-T) sensitivities for 99th percentile rainfall (A,C) and 99.9th percentile rainfall (B,D) with observed temperatures (A,B) and with corrected temperatures for cloud-effects (C,D).

References:

Tian, B., Chen, H., Yin, J., Liao, Z., Li, N., & He, S. (2023). Global scaling of precipitation extremes using near-surface air temperature and dew point temperature. *Environmental Research Letters*, 18(3), 034016.

Hosseini-Moghari, S. M., Sun, S., Tang, Q., & Groisman, P. Y. (2022). Scaling of precipitation extremes with temperature in China's mainland: Evaluation of satellite precipitation data. *Journal of Hydrology*, 606, 127391.

Li, X., Wang, T., Zhou, Z., Su, J., & Yang, D. (2023). Seasonal characteristics and spatio-temporal variations of the extreme precipitation-air temperature relationship across China. *Environmental Research Letters*, 18(5), 054022.

Pfahl, S., O'Gorman, P. A., & Fischer, E. M. (2017). Understanding the regional pattern of projected future changes in extreme precipitation. *Nature Climate Change*, 7(6), 423-427.

Gu, L., Yin, J., Gentine, P., Wang, H. M., Slater, L. J., Sullivan, S. C., ... & Guo, S. (2023). Large anomalies in future extreme precipitation sensitivity driven by atmospheric dynamics. *Nature Communications*, 14(1), 3197.

Reviewer #2 (Remarks to the Author):

This study explores the climate-scale cloud radiative effects on surface temperature during rainfall and quantifies their impacts through a thermodynamically constrained surface energy balance model. The study is an extension of the authors' previous works about estimate of heat flux based on a thermodynamically constrained energy balance model, and the cloud-radiative effect on surface temperatures and turbulent fluxes over land (Ghausi et al. 2023). The major results suggest that the projected intensification of extreme rainfall with temperature in tropics is closer to the ideal pattern of Clausius-Clapeyron rate when the cloud radiative effects are removed. While the overall results do provide certain evidence to support the explanation of the inconsistent relationship between temperature increase and extreme rainfall increase in observations, we have concerns about applying the thermodynamically constrained energy balance model to daily data at finer grid scale to seek relation between surface temperature (with and without cloud effect) and extreme rainfall (P95).

We thank the reviewer for investing their time and effort in reviewing our manuscript and for the thoughtful assessment of our work. The reviewer has raised some concerns about applying the thermodynamically constrained model at daily timescale. We have now performed additional evaluation to justify our approach. The analysis is extended to oceans as well (as suggested by the reviewer). The manuscript is accordingly revised with the point-by-point response mentioned in this document. The reviewer comments are shown in Black and our responses are shown in Blue. All the reviewer comments have been marked for reference as R2_M1, R2_M2 respectively. The reply is composed of a "response" to the comment and "action" taken in the main manuscript/supplementary file.

R2_M1: Although the authors claim that the approach have already been evaluated against observations and showed excellent agreement in reproducing turbulent fluxes and surface temperatures (Kleidon & Renner 2017; Ghausi et al., 2023), it appears that these evaluations are done based on data at longer time scale not at daily time scale. If true, there is no guarantee that the approach works on daily time scale because the simple energy balance may not hold in local convective processes for extreme rainfall. This is shown in the highly scattered EP-T relation in Fig. 3.

Response:

Thanks for the comment. The reviewer correctly points out that the evaluation in the previous studies (Kleidon & Renner 2017 and Ghausi et al., 2023) were done at coarser temporal scales. In Ghausi et al., (2022), it was performed at daily time-scale but was limited to a small region. We have now added here the global evaluation of this approach at the daily scale.

Fig. R1A (below) shows the comparison between the daily values of estimated surface temperatures from the thermodynamically constrained surface energy balance model with observations of surface temperatures derived from NASA-CERES dataset. Fig. R1B shows the map of root mean squared error (RMSE) between estimated surface temperature and observations from NASA-CERES, computed for the daily timeseries at each grid. While some biases exist, our approach reasonably captures the day-to-day variability in temperatures with

mean RMSE of 3.8K. We have further added the comparison between ERA-5 reanalysis data and NASA-CERES observations (Fig. R1C and R1D). We do this to highlight that residual errors in our model estimates are comparable and only slightly larger than those between the observations and reanalysis data.

Action:

Figure R1 is added to the supplementary information as Fig. S4. Text is added in the revised manuscript (lines 187-192).

Figure R1: (A) Comparison of daily estimated surface temperatures using the thermodynamically constrained surface energy balance model against observations from the NASA-CERES dataset. (B) Global map of root mean squared error (RMSE) between the estimated surface temperatures and NASA-CERES observations, computed for the daily timeseries at each grid point. (C, D) same as (A, B) but for the comparison between ERA-5 reanalysis data and NASA-CERES observations.

R2_M2: In fact, there is no physical base to expect EP-T to follow thermodynamic relation. There are few studies to suggest the P-T relation slower than the q-T relation in global energy balance condition, but no existing theories explain the EP-T relation. Even the q-T relation varies in different climate regimes (e.g., super greenhouse effect over the tropical warm pool). Therefore, I would expect to see some justification or evaluation of the thermodynamically constrained energy balance approach. Such an evaluation requires an estimate of quality of daily variables of solar and terrestrial radiative fluxes at surface, the surface temperature (that is different from near surface air temperature), and turbulent fluxes.

Response: We thank reviewer for the comment. We agree with the reviewer that in the global energy balance conditions, mean precipitation increases at rates much slower than q-T

relation due to the energetic constraints. The thermodynamically constrained approach used in our study was able to analytically reproduce these sensitivities as well (see Kleidon et al., 2013). However, the primary physical basis for rainfall extremes to follow the thermodynamic relation comes from the Clausius-Clapeyron equation that suggest atmospheric moisture to increase at the rate of 7%/K (Trenberth et al., 2003) at the assumption of small changes in relative humidity (Held & Soden, 2006). Climate models have shown rainfall extremes to increase at higher rates compared to mean rainfall but with regional variability as also mentioned by the reviewer. The deviations from the thermodynamic relation at regional scales have been attributed to the dynamic factors such as the strength of vertical velocity and moist convection (O’Gorman & Schneider, 2009; Pfahl et al., 2017; Gu et al., 2023).

To justify our thermodynamically constrained approach we would like it to support it with two lines of evaluation. First, we show that our thermodynamically constrained surface energy balance model effectively captures the observed day-to-day sensitivity of surface temperatures to changes in cloud-cover and comparable to that by ERA-5 reanalysis data. Secondly, we show that the regional deviations in EP-T sensitivities from the thermodynamic relation relates to the dynamics characterized by changes in vertical pressure velocities and these affects are very well captured in the changes in cloud base and cloud top temperatures.

- 1) We estimated the first-order sensitivity of daily surface temperature to changes in cloud cover at each grid point using temperatures derived from CERES, our thermodynamically constrained energy balance model, and ERA-5 surface temperatures. Fig. R2A compares the sensitivities of the temperatures estimated by our model with those derived from CERES, where each black dot represents the sensitivity at a specific grid point. Similarly, Fig. R2B compares the cloud-cover sensitivities of ERA-5 surface temperatures with those of CERES. Our results demonstrate that our model accurately captures the observed sensitivity from CERES with R^2 of 0.93 (Fig. R2A), and the residual errors were comparable to those in the sensitivities of ERA-5 and observations (Fig. R2B).
- 2) To assess the impact of dynamic changes on regional variations in EP-T sensitivities, we analysed standardized anomalies of vertical pressure velocity (w) on extreme rainfall days at each grid point. Vertical pressure velocity data at 650 hPa was taken from ERA-5 reanalysis. We plotted these anomalies against the spatial deviation in "clear-sky" EP-T sensitivities from CC rates (Fig. R3A, C). Our analysis revealed a positive relationship, indicating that the stronger anomalies in vertical pressure velocity led to positive deviations in sensitivities from the CC rate, while lower anomalies correspond to negative deviations. These findings are consistent with what have been reported by other studies as well (Pfahl et al., 2017; Gu et al., 2023). Note that, this relationship become clearer when the grids between 30°N and 30°S are isolated. This is the same region where the dynamic effects of changes in vertical velocities are most significant, as shown by Gu et al. (2023). We then show that these variations in vertical pressure velocities are closely linked to updrafts within clouds and are very well captured in the CERES cloud-base and cloud-top temperature data. Condensational heating generates power within the clouds to drive vertical motion and dehumidify the atmosphere. The more power clouds generate, the deeper the convection and the higher the rainfall. To analyse these differences in cloud updrafts

with increasing vertical velocities, we used the difference between cloud base and cloud top temperatures (obtained from NASA-CERES observations) as a proxy for moist convection within clouds. The stronger the updrafts lead to a greater temperature difference between the cloud base and cloud top. Fig. R3B, D demonstrate that the spatial deviation in EP-T sensitivities exhibits a clear, monotonic increasing relationship with cloud temperature differences as well. This relates back to the reviewer's point that the EP-T relation can deviate from the thermodynamic scaling at regional scales. We show that these changes can be attributed to changes in the atmospheric dynamics.

Action: Discussion is added in the main text (lines 187-192 and 269-282). Fig. R2 and Fig. R3 (below) are added as new supplementary figures Fig. S5 and Fig. S14.

Figure R2: (A) Comparison of the first-order sensitivity of daily surface temperatures to changes in cloud cover, estimated by the thermodynamically constrained energy balance model, with those derived from NASA-CERES observations. Each black dot represents the sensitivity at each grid point over land. (B) Same as (A) but for the comparison of cloud-cover sensitivities of ERA-5 surface temperatures with those derived from CERES observations.

Figure R3: Deviation in EP-T sensitivities (P95) from CC rate (7%/°C) after correcting for cloud-effects as a function of (A) standardized anomalies in vertical pressure velocity at 650 hpa on extreme rainfall (P95) days and (B) standardized anomalies in the temperature difference between cloud-base and cloud-top on extreme rainfall days (P95). (C, D) same as (A, B) but for grids between 30S and 30N.

R2_M3: A related question is about the change of water cycle with environment warming. For completeness, it is desirable to see the same analysis over oceans. The SST, air-sea fluxes, and rainfall are all available over oceans. It is physically more suitable and practically easier to apply the method to search EP-T relation over oceans.

Response: As per reviewer' suggestion, we have now extended the analysis to oceans as well. We used rainfall-temperature data from ERA-5 reanalysis, together with the radiative fluxes from NASA-CERES to estimate the EP-T sensitivities with and without the effect of clouds.

Observed EP-T scaling over oceans is shown in Fig. R4C (below). We find that negative scaling dominates across the tropical oceans. These estimates are consistent with what have been reported by other studies (Wang et al., 2017; Tian et al., 2023; Gu et al., 2023). To estimate the effect of cloud-cooling over oceans, we used our thermodynamically constrained surface energy balance forced with "all-sky" and "clear-sky" radiative fluxes. The evaluation of the estimated daily surface temperatures with observations from CERES over oceans is shown in Fig. R4 A, B. We find that the residual errors were higher than that compared to land. This can be attributed to the surface heat -storage effect within the oceans as well as the oceanic heat transport that remain unaccounted for in our approach (more details in Dhara et al., 2016 & Kleidon & Renner, 2017). However, this will likely lead to a systematic bias (Dhara et al., 2016)

which should not affect the difference between the “clear-sky” and “all-sky” temperatures which is quantified as cloud-radiative cooling. Thus, our approach can be used to diagnose the effect of clouds on ocean temperatures as well.

EP-T scaling over oceans after removing the effect of clouds are shown in Fig. R4D. The scaling became positive across the tropical oceans. These estimates were consistent with what have been reported using dew-point temperatures as a scaling variable over oceans (Tian et al., 2023) as well as the climate model projections (Chang et al., 2022). However, it should also be noted that apart from clouds, ocean surface temperatures can also cool due to wind-induced upwelling of deep sea-water, turbulent mixing of the oceanic layers and heat extraction by storm during the rainfall event. So, while we show that the cloud-cooling effects explains negative sensitivities in tropical oceans as well, we note that the net estimated cooling of surface temperature over oceans may be underestimated by accounting for the effect of clouds alone. A detailed analysis to quantify the cooling effect of storm over ocean will require a different formulation of surface energy-balance model that explicitly accounts for the heat-storage and wind mixing effects over the ocean and remain a potential area for future research.

Action: Text is added in lines 283-292 of the revised manuscript. Figure R4 (below) is added as new supplementary figure Fig. S12.

Figure R4: (A) Comparison of estimated daily sea surface temperatures using the energy balance model with temperatures derived from NASA-CERES observations. Root mean squared error (RMSE) between daily estimated and observed temperatures from CERES across all the oceanic grids. Global map of extreme

precipitation-temperature sensitivities over oceanic grids with (C) observed temperatures and (D) temperatures corrected for cloud-effects.

R2_M4: Besides the above major concerns, we wonder whether the cloud cooling effect is the most important factor influencing EP-T. The authors mentioned some other explanations in the text. Is it possible to isolate the EP-T under greenhouse warming in the current climate?

Response: We thank reviewer for the comment. In this study, we show that the negative extreme precipitation-temperature sensitivities in observations across the tropics arises predominantly due to the cloud-cooling effect. However, it necessarily does not mean that cloud-cooling effect is the most important factor influencing other aspects of EP-T sensitivities as well, for e.g., the regional variability and deviations from CC scaling. While, we show in figure 4 (main manuscript) that the regional variability in the extreme precipitation-temperature (EP-T) sensitivities is reduced after removing the cloud radiative effect from temperatures, there still exist regional variations and deviations from CC rate in the estimated “clear-sky” sensitivities. Studies have shown that these deviations arise due to the dynamic component characterized by changes in vertical pressure velocities (Pfahl et al., 2017; Gu et al., 2023). We have now also demonstrated in Fig. R2A (above) that regional deviations from CC scaling correlates with changes in vertical pressure velocities and moist updrafts within clouds. Other aspect is the existence of super CC rates which cannot be explained by effect of clouds alone and are related to the strengthening of moist convections.

To isolate EP-T response under greenhouse warming in the current climate is not possible using our thermodynamic energy balance model as it does not simulate precipitation directly but rather the surface temperatures and turbulent fluxes of sensible and latent heat associated with changes in radiative forcings. This model has been used to obtain sensitivities of global mean evaporation and have been found to reproduce the sensitivities from climate model simulations very well (see Kleidon & Renner 2013, Kleidon et al., 2014). However, we note that other studies using optimal fingerprint techniques with climate model runs have shown that observed intensification of extreme rainfall is attributable to greenhouse induced warming (Zhang et al., 2013; Paik et al., 2020).

Action: Text is added in lines 269-282.

Overall, the objectives and findings of this study are quite interesting but the applicability of the method needs be justified to ensure the major conclusions are reasonable. In addition, some statistics, physical discussions, and method descriptions are unclear and must be revised. Therefore, I recommend that this article be considered for publication after the above concerns are adequately addressed.

Thank you, these are helpful comments and we hope we have now addressed your concerns adequately.

Specific comments:

1. The extreme precipitation-temperature (EP-T) sensitivities, scaling rate, and quantile regression should be briefly introduced/defined in the main text, although they are described in the section of Data and Methods.

Response: Thanks for the suggestion. We have made this change.

Action: Text is added in lines 94-97 of the revised manuscript.

2. Please add more physical discussion about possible mechanisms causing the different effects among the tropics, mid-, and high-latitudes.

Response: The suggested change is made.

Action: Text is accordingly modified in the discussion section.

3. Please highlight the potential applications of the findings in this study or the expansion of our physical understanding in the last 2-3 paragraphs. They are unclear in the current manuscript.

Response: Thanks for the suggestion. We have now added the potential applications of the findings in this study in the last paragraphs.

Action: Text is accordingly modified in the discussion section.

4. The “significant results” (e.g., lines 129, 244, 274, Fig 4, Fig S3, S4, S7) should be examined by a statistical significance test to precisely mention and compare their significant differences or regressions.

Response: Thank you for the suggestion.

For the relationship between precipitation and cloud radiative effects, as well as changes in temperature, we have now reported the p-value for the estimated slope coefficient to demonstrate the statistical significance of the results.

For the variability in precipitation-temperature sensitivities, we performed an F-test, which compares the variances and determine if the variability between two datasets is significantly different. The F-test revealed that the reduction in variability in extreme-precipitation-temperature sensitivities after removing the cloud radiative effects was statistically significant ($p < 0.0001$).

Action: Text is added in lines 143 and 254-257 of the revised manuscript.

5. Please add a brief description of “the bin-shifting effect” in line 152 to the main text or an example figure in the supplement.

Response: Thanks for the suggestion. We have now added a description about the bin-shifting effect in the main text.

Action: Text is added in lines 165-167 of the revised manuscript.

6. Line 199: Fig S5 should be Fig S6.

Response: Thanks for pointing this out. The suggested change is made.

7. CPC should be in full name when it first appears.

Response: The suggested change is made.

8. Each possible mechanism mentioned in lines 235-237 should have corresponding references or brief descriptions.

Response: The suggested change is made.

9. Please ensure that either °C or K is used consistently throughout the manuscript.

Response: The suggested change is made. We have used °C at all the places where EP-T sensitivity is discussed.

10. Lines 240-242: It should be also highlighted in the Introduction.

Response: The suggested change is made.

11. Line 246: has

Action: Changed

12. Line 260: remains

Action: Changed

13. Line 265: suggest

Action: Changed

14. Line 272: highlight

Action: Changed

15. Line 274: show

Action: Changed

16. Line 283: induced

Action: Changed

17. Line 285 shows

Action: Changed

18. Please check and add many missing references, such as Emory & Brown., 2005; Lenderink et al., 2010; Dhara et al., 2016; Kleidon & Renner 2017; Wasko et al., 2018; Wasko & Nathan 2019; Tian et al., 2023a, b ...etc.

Action: Thanks for pointing this out. The missing citations have been added.

References:

Chang, M., Liu, B., Wang, B., Martinez-Villalobos, C., Ren, G., & Zhou, T. (2022). Understanding future increases in precipitation extremes in global land monsoon regions. *Journal of Climate*, 35(6), 1839-1851.

Kleidon, A. and Renner, M.: A simple explanation for the sensitivity of the hydrologic cycle to surface temperature and solar radiation and its implications for global climate change, *Earth Syst. Dynam.*, 4, 455–465, <https://doi.org/10.5194/esd-4-455-2013>, 2013.

Kleidon, A., Kravitz, B., & Renner, M. (2015). The hydrological sensitivity to global warming and solar geoengineering derived from thermodynamic constraints. *Geophysical Research Letters*, 42(1), 138-144.

Zhang, X., Wan, H., Zwiers, F. W., Hegerl, G. C., & Min, S. K. (2013). Attributing intensification of precipitation extremes to human influence. *Geophysical Research Letters*, 40(19), 5252-5257.

Paik, S., Min, S. K., Zhang, X., Donat, M. G., King, A. D., & Sun, Q. (2020). Determining the anthropogenic greenhouse gas contribution to the observed intensification of extreme precipitation. *Geophysical Research Letters*, 47(12), e2019GL086875.

We thank the reviewers for investing their time and effort in reviewing our manuscript and for the positive and thoughtful assessment of our work. The manuscript is accordingly revised as per reviewer's comments and suggestions, with the point-by-point response mentioned in this document. The reviewer comments are shown in Black and our responses are shown in Blue. All the reviewer comments have been marked for reference as R3_M1, R3_M2 respectively. The reply is composed of a "response" to the comment and "action" taken in the main manuscript/supplementary file.

The observed scaling relationship between surface temperature and extreme precipitation daily amount is revisited at global scale. The temperature is corrected for, thanks to a surface budget equation, to remove the cooling effect of the cloudiness associated with the extreme precipitation event. The resulting correction improves the scaling, in particular in the tropics.

The authors should be commended for connecting the cloud radiative effect of the precipitating system to the surface temperature in the scaling relationship. This is a timely addition to the debate on the usefulness of such scaling laws for assessing climate models predictions of future extreme precipitation.

Thank you for the positive feedback.

While this reviewer is rather in favor of publication of this paper, it will increase the impact of this work if the paper would include a discussion about:

R3_M1: - physical model behind the scaling (e.g. Muller and Takayabu, 2020) and how the corrected temperature actually enters the scaling equation?

Response: This is a very good suggestion. We have now included a brief background on the physical model behind scaling in the methods section and related how corrected temperature enters this equation. Discussion about this have also been added in the main text.

Action: Text is added in lines 388-409 of the revised manuscript.

R3_M2: alternative to this correction. In particular using time lagged temperature instead of contemporaneous ones over both land (Bao et al., 2017) and ocean (De Meyer and Roca, 2021) and how the present correction does a better job at characterizing the surface condition that yield to the extreme precipitation event?

Response: Thanks for the comment. The reviewer correctly points out that studies have tried to minimize the cooling effect of rain by using time-lagged values of surface temperatures. While this leads to some improvement in the scaling, it has been shown that it does not entirely remove the negative scaling over land (Bao et al., 2017).

To test this in our study, we repeated the analysis by using 1-day lagged temperature to estimate the extreme precipitation scaling rates over land. The results are presented in Fig. R1 (below). The box plots in Fig. R1 (bottom-row) compare the sensitivities obtained using observed (red), time-lagged (orange) and corrected temperatures after removing the cloud-effects (blue). Although, the sensitivities improved after using lagged temperatures, they were

still negative across most regions of the tropics. Similar pattern was found in the residual error between observations and fitted quantile regression. The residuals were lower for time-lagged temperatures compared to observation but higher than those obtained after removing the cloud-radiative effects. This shows that while using lagged-temperatures minimize cooling for some events, it does not entirely eliminate the cooling effect particularly where the time-scale of cooling is longer than a day or few like the monsoon-systems (Ghausi et al., 2022). Secondly, the cloud-formation does not necessarily imply the storm-event occurrence. As a result, any delay between cloud-formation and storm will not be accounted for while using lagged temperatures but will be effectively removed using our approach.

Action: Text is added in lines 330-334 of the revised manuscript. Figure R1 (below) is added as a new supplementary figure S15.

Figure R1: (Top row) Global map of extreme precipitation-temperature sensitivities estimated using quantile regression with (A) observed temperatures, (B) 1-day time lagged temperatures and (C) temperatures corrected for cloud-radiative effects. (Bottom row-D) Comparison of EP-T scaling rates estimated using observed temperatures (red), time-lagged temperatures (orange) and with temperatures corrected for the cloud-cooling effect (blue) for tropics, mid-latitudes and high-latitudes. (E) same as (D) but for the residuals between observations and fitted quantile regression

On a miscellaneous note, I would suggest to state early in the manuscript the source of the precip data and the scale (daily) as well as the scale upon which the cloud radiative effect is computed.

Response: We thank the reviewer for the suggestion. We have now added the information about the precipitation data sources and the temporal scales of the analysis at the end of the introduction section.

Action: Text is added in lines 80-83 of the revised manuscript.

References added:

Bao, J., Sherwood, S., Alexander, L. et al. Future increases in extreme precipitation exceed observed scaling rates. *Nature Clim Change* 7, 128–132 (2017). <https://doi.org/10.1038/nclimate3201>

De Meyer, V. and R. Roca, 2021: Thermodynamic scaling of extreme daily precipitation over the tropical ocean from satellite observations. *J. Meteor. Soc. Japan*, 99, 423-436.

Muller, C. J., & Takayabu, Y. (2020). Response of precipitation extremes to warming: what have we learned from theory and idealized cloud-resolving simulations, and what remains to be learned? *Environmental Research Letters*. <https://doi.org/10.1088/1748-9326/ab7130>

We thank the reviewer for investing their time and effort in reviewing our manuscript and for the positive and thoughtful assessment of our work. The manuscript is accordingly revised as per reviewer's comments and suggestions, with the point-by-point response mentioned in this document. The reviewer comments are shown in Black and our responses are shown in Blue. All the reviewer comments have been marked for reference as R1_C1, R1_C2 respectively. The reply is composed of a "response" to the comment and "action" taken in the main manuscript/supplementary file.

Reviewer #1 (Remarks to the Author):

R1.C1: The authors address most of mine concerns, this is good. However, I still have some doubts regarding the scaling patterns across India and southeast China. The authors argue that 'different regimes of how CRE scales with rainfall may likely relate to the contribution from large scale and convective rainfall events.' Since it is tropics, I would like to believe large scale circulation plays a minor role while small scale circulation and convection can be more important. Moreover, is this phenomenon can be related to 'monsoons'? However, even during monsoon seasons, strong precipitation across the India and southeast China can be the same... Can the authors discuss or analyze more details about how the monsoon affect the different scaling across India, southeast China, and Indonesia, respectively?

Response: Thanks for the comment. We agree with the reviewer that in the tropics small-scale circulation and convection will play a major role. We believe that different regime of how cloud radiative effects (CRE) scales with rainfall mainly relate to spatial structure of cloud-cover and its areal spread. NASA-CERES data only provides a spatial CRE across a $1^{\circ} \times 1^{\circ}$ grid resolution and might not be able to truly represent the small-scale dynamics of cloud-cover. We believe this is an important effect but will ideally require Flux-tower observations across these regions to diagnose this at local scales and have been kept out of scope for present study.

About the role of monsoon: this is a very good point. We looked more into how monsoon can affect scaling over these regions. We isolated the grid across south Asia into three regions: Indian region, Eastern China and South Eastern countries Thailand, Laos and Vietnam. While the scaling after removing the cloud-radiative effects is positive and quite uniform everywhere, the eastern humid region of China also showed positive scaling with observed temperatures (Fig. R1, below). We have already attributed it to how CRE scales with precipitation over these regions (Supp. Fig. S11).

We then also looked at the seasonal cycles of precipitation, temperature and cloud-radiative effects over these regions. We find that the Indian and South east asia region shows a strong skewness in the annual temperature variations (due to strong monsoon seasonality) leading to high pre-monsoon temperatures (extreme left and right column in figure R1), while this effect is absent across the Eastern China region which despite having a monsoonal seasonality also receives some rainfall throughout the year. The presence of high pre-monsoon temperatures has already been argued to exacerbate negative scaling by intensifying the bin-shifting effects (Ali & Mishra, 2017; Ghausi et al., 2022; Dash & Maity, 2022) and this effect was found to be absent across Eastern China region.

Action: Text is added in lines 221-225 of the revised manuscript.

Figure 1: (top row) Maps of Extreme precipitation sensitivities before and after removing the radiative effect of clouds respectively. (Bottom row) Seasonal cycles of precipitation, temperature and cloud-radiative effects over (left) Indian monsoon region, (middle) Eastern China region and (right) South Eastern countries: Thailand, Laos and Vietnam.

R1.C2: Secondly, I still consider that in the main text, dividing all scaling patterns into tropics, mid-latitude, high latitudes are not that appropriate? Maybe it's not a big problem, but since the authors have already global maps, why they always tend to group the results into 3 region boxes? Since there are discernable zonal differences of the scaling rates.

Response: Thanks for the comment. We totally agree with the reviewer that there are discernable zonal differences in the scaling rates visible from the maps. However, we chose to group the results into three regional categories (tropics, mid-latitudes, and high latitudes) for two main reasons:

- 1) Reduce variability: Displaying scaling curves globally, especially for each land grid, introduces substantial interzonal variability. By categorizing into tropics, mid-latitudes, and high latitudes, we significantly reduce this variability, resulting in clearer and more interpretable visualizations.
- 2) Maintain consistency with other studies: Monotonic decreasing scaling in the tropics, hook-shaped scaling in the mid-latitudes and monotonic increasing scaling at high latitudes have been widely argued in literature (Utsumi et al., 2011; Yin et al., 2018). Recent studies that address this issue have divided the scaling in similar latitudinal bins (Tian et al., 2023). To maintain the consistency of results with other studies we adopted a three-region approach.

R1.C3: Further, does the aridity can affect the scaling rates or cloud effects? For example, how does the observed scaling rates varies and how cloud cooling affects the eastern North America (belongs to humid areas) and the western North America (belongs arid areas), respectively?

Response: This is a very good point. First, we would like to point out that aridity is closely related to cloud-cover. Temperature variations across aridity gradient have been shown to be strongly modulated by cloud-radiative effects (Ghausi et al., 2023). This also makes cloud-radiative effects to differ considerably across dry and humid regions.

Secondly, Negative scaling has been widely argued to be due to moisture limitations (higher aridity). While we show that cloud radiative effects can clearly explain negative scaling across most global land regions. There still exist some dry regions where negative scaling persists even after removing the cloud-cooling effects. This can be seen in arid region of western North America and Sahara. This means that over these regions the moisture limitation (aridity) may be the major driver of negative scaling compared to the effect of clouds. We have already noted this point in lines (225-228) of the manuscript, but agree that a more comprehensive analysis of aridity's role could strengthen the discussion.

To first demonstrate specifically about North America that reviewer pointed out, we show in figure R2, the maps of aridity, net cloud radiative effects, observed and clear-sky scaling over this region. We can clearly see that in the arid western region (figure R2 A), the cloud radiative effects are much less strong (Fig. R2 B). Initially the negative precipitation scaling persists across the whole regions (Fig. R2 C). However, after removing the cloud radiative effects, scaling across most of the eastern humid regions became positive while some negative scaling still remained over western arid region (Fig. R2 D). We believe that this scaling is due to moisture availability limitations and cannot be explained by the effect of clouds.

To demonstrate this effect at the global scale, we first show in figure R3A that the net cloud radiative effects become weaker as we go towards drier regions indicating less clouds. The effect of clouds in modulating surface temperatures across the aridity gradient have been described in detail in Ghausi et al., (2023). Secondly, we show that clear-sky extreme precipitation-temperature sensitivities also shows a first-order reduction as we go towards drier regions (Fig. R2B). This shows that moisture availability limitation could still play an important role in explaining negative/reduced scaling over dry regions.

Action: Text is added in lines 229-233 of the revised manuscript. Fig. R3 is added as new supplementary figure S16.

Figure R2: Maps over North America grids for (A) Budyko Aridity Index (defined as the ratio between mean net radiation and the energy equivalent to mean annual precipitation), where higher values indicate more arid conditions. (B) Net Cloud Radiative Effect (W/m^2). (C) Observed extreme precipitation-temperature Scaling (D) Extreme precipitation-temperature scaling after removing the effect of clouds.

Figure R3: Variation along the Budyko aridity index (defined as the ratio between mean net radiation and the energy equivalent to mean annual precipitation), where higher values indicate more arid conditions of (A) Net Cloud radiative effects (W/m^2) and (B) Extreme precipitation-temperature scaling after removing the effect of clouds.

R1.C4: Finally, it seems that topography can also affect scaling (e.g., different scaling between the Himalaya Mountain regions and southeast Asia). Although it is not the target of this research, the authors can have a short discussion.

Response: Thanks for the comment. The topography indeed plays an important role in affecting the scaling due to several reasons:

- 1) The presence of orographic lifting can substantially deviate precipitation sensitivities to follow the thermodynamic scaling and the dynamic factors become much more important over these regions (Moustakis et al., 2020).

- 2) Due to changes in the local radiative environment at high elevation regions, the sensitivity of surface temperatures to changes in cloud-cover can vary significantly compared to low elevation regions. For instance, high elevation regions have been shown to be much more sensitive to changes in the atmospheric emissivity and downwelling longwave radiation which can influence the magnitude of net cloud radiative effects associated with rainfall events.

Action: We have added a short discussion about it in the main text. Text added in lines 226-228 and 317.

Reviewer #2 (Remarks to the Author):

The revised manuscript addresses all my comments adequately. I recommend the manuscript be accepted for publication after some very minor changes as stated below. I do not need to see the revision again.

R2.C1: The descriptions about figure S4ab are unclear and confusing. I do not understand NASA-CERES temperature. From the “dataset used”, I see CPC temperature but no NASA-CERES temperature.

Response: We apologize for not being clear. From NASA-CERES temperatures, we refer to the surface temperatures diagnosed using upwelling longwave equation from NASA-CERES dataset. We have now clarified it in the caption.

Action: Caption of figure S4a, b is revised.

R2.C2: In LINE 465, equation (S3) is referred but nonexistent. Please check equation numbers in the text.

Response: Thanks for pointing this out. We have made the correction.

Action: We have corrected the text.